

# Determinants of thermal regime influence of small dams

André Chandesris [1], Kris Van Looy [2], Yves Souchon [1]

[1] River Hydro-Ecology Lab, National Research Institute of Science and Technology for Environment and Agriculture, UR Riverly, Lyon, France

[2] OVAM, Stationsstraat 110, 2800 Mechelen, Belgium

*Correspondence to*: A. Chandesris (andre.chandesris@irstea.fr)

**Abstract.**

The purpose of this study was to quantify the downstream impacts of different types of small dams on summer water temperature in lowland streams. We examined: (i) temperature regime upstream and downstream dams of different structural

characteristics; (ii) relationships between stream temperature anomalies and climatic variables, watershed area, dam height, impoundment length and surface and residence time; (iii) the more significant variables explaining the different thermal behaviours, in order to account for dam diversity and functioning in future regional stream temperature models.

Water temperature loggers were installed upstream and downstream  11 dams in the Bresse Region (France) and monitored at 30 min intervals during summer period (June to September), from 2009 to 2016 depending on the sites (2 sites were

monitored during 2 summers, others only 1 summer, resulting in 13 time-series), with the opportunity to compare cold and hot summers.

The small dams altered the downstream thermal regime for 23 % of the time-series with a > 1°C elevation of the maximum daily temperature; for 77 % the range was in between -1°C and +1°C. The mean increase of the minimum daily temperature was 1°C, with 85 % of the time-series showing an increase > 0.5 °C.

The sites are grouped in three main types with specific responses of different temperature variables (maximum daily temperature (T max), minimum daily temperature (T min) and daily temperature amplitude). Two main types of impact were identified: an increase in the daily minimum temperatures associated with stability and even a slight reduction of the daily maximum temperatures for impoundments of low volume (residence time shorter than 0.7 day and an impoundment surface area smaller than 35 000 m$^2$); and an increase of the daily minimum and maximum temperatures in the same orders of

magnitude when the surface area of the impoundment is larger than 35 000 m$^2$ and the residence time is longer than 0.7 day. This increase can reach 2.4°C at certain structures and could impact the structure of aquatic communities and the functioning of the aquatic ecosystem.  These determinants are candidate to generalize results, but this would necessitate the gathering of more precise information than the current dam descriptors in public databases.

**Keywords:**  Stream, water temperature, summer thermal regime, small dam

## 1    Introduction

### 1.1    The temperature is a master physical variable in streams

 The water temperature governs the geographical range, condition and physiology of aquatic organisms (Allan and Castillo, 2007) and the stream metabolism (Bernhardt et al., 2018; Brown et al., 2004). As ectotherms, aquatic organisms are very

sensitive to ambient water temperature and to its alteration, especially in the vicinity of their upper thermal temperature tolerance (Brett, 1979; Coutant, 1987; McCullough et al., 2009 for Coldwater fish review; Souchon and Tissot, 2012 for





European non salmonid fish review). The temperature governs the life history of invertebrates by affecting egg development, fecundity, dormancy, growth, maturation, voltinism, and emergence (Rader et al., 2007 in Ellis and Jones, 2013). Understanding the river thermal regime is therefore crucial to understand ecological functioning (Hester et al., 2009),
particularly in an era of global warming (IPCC, 2007 and 2013) and numerous ecological changes (Woodward et al., 2010).

### 1.2    The different water temperature drivers

Major natural drivers of water temperature are (1) climate, i.e. solar radiation, air temperature, wind, precipitations, upstream water temperature, (2) topography, i.e. stream orientation, stream shading by surrounding vegetation, (3) stream bed, i.e. hyporheic exchanges, groundwater input, and (4) stream discharge (Caissie et al., 2006). These physical variables responsible
for the balances of heat fluxes governing the temperature of streams (Caissie, 2006; Hannah et al., 2004 and 2008; Kelleher et al., 2012; Mohseni et al., 1998; Webb et al., 2008) can be used to identify environmental determinants for a given site. During summer, the factors leading to warming are: (i) the input of heat from upstream (depending on discharge and water temperature); (ii) direct and indirect solar radiation dominated by infrared radiations; (iii) air-water conduction (convective heat flux or sensible heat); and (iv) bed conduction. The factors leading to cooling are: (i) longwave radiation emitted by the
water surface; (ii) latent heat; (iii) the influx of groundwater. The thermal regime could be altered by different anthropogenic structures, punctual in the case of dams or more spatially distributed in the case of riparian vegetation clearings. Their impacts vary in spatial and temporal scope, depending on relative size effects of stream (headwater to river) versus human features (i.e. reservoir volume; length of vegetation clearings). Dams are susceptible to modify the thermal balance of the stream by storing volumes of water, and by increasing the contact surface of a stream with the atmosphere.

### 55    1.3    Large dam effects

The Hester and Doyle (2001) literature review reveals that the cooling effect of large dams (> 15 m) with stratified waters in impoundment and hypolimnion occurring downstream is the most described worldwide. The Serial Discontinuity Concept (SDC, Ward and Stanford, 1983) is largely based on this property of water cooling by large stratified impoundments which alter longitudinal downstream water temperature pattern tens of km depending on dam characteristics, flow regime, river
physical characteristics and downstream inputs of lakes, groundwater, and tributaries (Olden and Naiman, 2010; Ellis and Jones, 2013 for a review). In addition, Ward and Stanford (1983) predicted that dams in headwaters might not alter the natural temperature range, with the assumption that canopy and springs or groundwater influx can buffer annual temperature variations. Furthermore, SDC predictions mentioned summer water temperature warming downstream of surface release reservoirs (O'Keeffe et al., 1990).

### 65    1.4    Small dam characteristics are not well established

But less is known about quantitative thermal impacts of small dams, especially run-of-the-river dams with few or no vertical thermal stratification and surface release (Cumming, 2004; Hayes et al., 2008). Due to the surface of the impoundment exposed to solar radiation and decreased flow velocity, they are expected to deliver downstream warmer water, contrary to large dams with cold hypolimnion release. Moreover, they have often been built for a long time for a variety of uses (mills,
irrigation, livestock watering, storm water management, esthetic lakes in the residential landscape, hydroelectricity or stabilization of the longitudinal profile following stream channelization and straightening). Contrary to large dams, their number, their spatial location and their characteristics are not well known or often very imprecise depending on nation databases. For example, International Commission of Large Dam (ICOLD, 2017) inventoried 59071 dams in 160 countries



with a height of 15 meters or greater from lowest foundation to crest or a dam between 5 meters and 15 meters impounding
more than 3 million cubic meters. The number of smaller dams could be several million in the world: by combining data
published in Messager et al. (2016), the number of impoundments with surface > 10 ha could be estimated around 2 710000
and there will exist still more impoundments with smaller surfaces. In France, the National Inventory of Dams and Weirs
database (ROE, sept 2017) maintained by the French Biodiversity Agency, inventories 96 222 hydraulics works crossing
streams and rivers. This corresponds to a density of 0.42 obstacles per km on a basis of 230 000 km streams with permanent
flow. Their complete characteristics have not yet been quantified, knowing that the height alone is not sufficient to
discriminate their effects (Poff and Hart, 2002). MBaka et al. (2015) proposed a definition for the different features,
considering Run-of-the River Dam (RRD) as the smaller ones (height not exceeding river bank elevation and water flows
over the dam), Small Weirs (SW) corresponding to heights around 5 m and small impoundment and Low Head Dams (LHD)
with heights comprised between 5 and 15 m and an impoundment. We studied dams < 5 m, called hereafter simply small
dams.

## 1.5     Small dam thermal effects

In their review, Hester and Doyle (2011) concluded that most typical human impacts including small dams alter stream or
river temperatures by about 5°C or less. M'Baka et al. (2015) in their global review of downstream effect of small
impoundments found very few articles dealing with temperature effects: on 43 sites, 25 % have been considered having a
temperature increase effect, 2 % a decrease effect and 73 % no change. Dripps et al. (2013) studying 3 residential artificial
headwater lakes (17 to 45 ha) on stream (low flow discharge 0.0024 to 0.0109 m$^3$/s) showed that they could increase summer
downstream temperature by as much 8.4°C and decrease diurnal variability by as much 3.9°C. Maxted et al. (2005) found
that impoundments (height < 5 m and surface < 1 ha) in rural catchments increased mean daily stream temperatures by 3.1-
6.6°C during the critical summer period, and temperature differences were three times higher than those in woody
catchments (0.8-2.0°C). Hayes et al. (2008) in the region of Great Laurentian Lakes measured a weak to null thermal effect
of low-head barriers (<0.5 m in height) built to prevent the upstream migration of sea lamprey *Petromyzon marinus*, but a
temperature elevation comprised between 0.0 to 5.6°C below small hydroelectric dams. Analyzing the thermal effects of
beaver dams, Weber et al. (2017) underlined the complexity and the diversity of the situations, some authors concluding in
support of little to no influence (Sigourney et al., 2006), and others measuring extreme temperature increase of 7°C in a
headwater passed through large (5 ha) beaver dam complexes (Margolis et al., 2001).
These different studies show the extreme variability of the situations and the difficulty to identify the master variables
governing the thermal regime, dam height alone appearing as a poor descriptor. Nevertheless several explanatory variables
have been identified: stream size, stream order, watershed surface and vegetation cover, climate context, geology and alluvial
aquifers, groundwater exchange, impoundment surface directly submitted to radiation, water residence time, and base flow
discharge. This variability is greater in headwaters due to the weak thermal inertia and great diversity of these waterbodies,
and also to heterogeneous effects with closed riparian canopy or aquifers. This is the reason why it seems preferable in a first
study to focus on the single effects of the impoundment immediately downstream the dam.

## 1.6     Objective of the study

The purpose of this study was to quantify the downstream impacts of different types of small dams on summer water
temperature in lowland streams, when it reaches maximal values liable to impact ecological functioning. We examined: (i)
temperature regime upstream and downstream dams of different structural characteristics; (ii) relationships between stream





temperature anomalies and climatic variables, watershed area, dam height, impoundment length and surface and water residence time; (iii) the more significant variables explaining the different thermal behaviors, in order to account for dam diversity and functioning in future regional stream temperature models.

## 2    Material and methods

### 2.1    Study area

The sector concerned is an alluvial lowland plain northeast of Lyon, between the Jura and the north Massif Central (Fig. 1), at altitudes between 170 and 320 m. The network of tributaries of the Saone of small and medium streams (Strahler order from 1 to 5) is affected by a general environment of agricultural (67.4% compared to the French average of 59.5 %) and urban pressures (7.2 % - French average 5.5 %) (UE-SOeS, CORINE Land Cover, 2012), characteristic of temperate European plain regions, with a dam and weir density of 0.64 features per km greater than the French average of 0.42 features per km (Référentiel national des Obstacles à l'Ecoulement, ROE, September 2017) on a basis of 230 000 km streams with permanent flow. The density of the stream network is comparable to that of the national average (0.4 km/km$^2$).

In the study area climate is cold continental, characterized by hot dry summers (average maximum temperature 25.8°C) and cold winters (average maximum temperature 5°C). Average annual precipitation for Bresse region is 900 mm. This region is distinguished climatically by maximum median air temperatures in July (period 1960-1990) exceeding 25.5°C, equivalent to those of the Mediterranean region and of southwest France (Wasson et al., 2002). Regionalized climate projections on the scale of France (Peings et al., 2012) showed that this region is susceptible to being affected by higher summer air temperatures, with increases in the region of 2 to 3°C for maximum daily temperatures. For scenario A1B (mean concentration of greenhouse gases), the estimation was more than ten additional days of heat waves by 2050.

### 2.2    Sampling sites

The dams in the study area are overflow structures and stand on the sites of former water mills, some of which are still used to produce energy. The sites monitored were dams with heights varying between 1.0 m and 2.4 m, with a length of backwater flow ranging from 280 m to 2 950 m, and an impoundment with volumes ranging from 1200 to 53000 m$^3$. The average residence times is calculated by the ratio impoundment volume (m$^3$) / per daily water flow volume (m$^3$/day). The daily water flow volume is reconstituted using hydrometric measurement sites (French database HYDRO) and weighted by a correction coefficient during low flow periods estimated on the basis of punctual gauging performed by the regional service responsible for hydrometry (Direction Regionale de l'Environnement, de l'Amenagement et du Logement, DREAL). These residence times vary from 0.1 to 8.4 days (Table 1).

The structures studied differ considerably in terms of the surface area of the impoundment upstream of the weir, the ratio between the volume of the impoundment to the discharge, expressed by residence time, and their position in the hydrographic network (Table 1). These variables govern: (i) the input of diurnal heat from solar radiation; (ii) the loss of nocturnal heat linked to evaporation, emitted radiation; (iii) the upstream permanent inflow of heat.

### 2.3    Temperature monitoring

The temperature sampling was performed in summer (from the end of June to the beginning of September) from 2009, by the local water management body (Syndicat Mixte Veyle Vivante).





The temperature recorders used (Hobo®, Onset Computer Application; accuracy +/- 0.54°C) were installed in the vicinity of each structure: upstream of the hydraulic effect of the dam and downstream (< 100 m) of the dam in the main flow of the channel, at depths between 20 and 50 cm.

The temperature was recorded at a time step of 30 minutes.  The temperature recorders are calibrated each year using the simple procedure named "ice bucket" method Dunham et al. (2005).

The summer climatic characteristics during monitoring showed a predominance of higher than normal air temperatures for these years, except for 2014 (only one site), which was colder with significantly higher precipitations. For the other years precipitations are close to the normal, except in 2009 and 2016, with lower quantities (Table 2).

**2.4   Data analysis**

To determine if the dams alter the temperature regime, the minimum, average and maximum temperatures and amplitudes were calculated for each full day recorded, and the median values were recorded for the period. The calculations of daily differences of maximum and minimum water temperatures were performed for each pair of upstream/downstream records, and the median of these differences over the recording period was calculated. In addition, different focus are provided to

analyze (i) the regression between upstream and downstream water temperature on a daily pattern; (ii) the relationships between air and water daily temperature during all the recording period; (iii) the differences of temporal pattern of warming between upstream and downstream dam on a three days length (iv) the dam thermal effect considering an arbitrary threshold of 22 °C, with a calculation of the number of days above this threshold.

Finally, we propose a classification of the observed thermal behavior in 3 groups, based on differences between upstream and

downstream dam daily maximum temperature, daily minimum temperature and daily amplitudes.

**2.5   PCA analysis**

In order to identify the characterization of the impacts of the different dams, a principal component analysis (PCA) was carried out using the software XLStat (ADDINSOFT™) on the water temperature variables: downstream / upstream difference of the maximum, average and minimum daily temperature and daily temperature amplitude. The physical

characteristics of the structures (Table 1) were used as illustrative variables to evaluate the correlations with the temperature variables.

**3       Results**

**3.1     General temperature patterns**

The general natural pattern of temperature variations during summer for monitored streams presents the following

characteristics (Fig. 2):
-   a classical daily (diel) variation of temperature (minimum in early morning, maximum in late evening),
-   periods of progressively increasing minimum daily temperature values (T min) and maximum daily temperature values (T max), sudden falls in temperature are registered during the monitoring, generally linked to precipitation events.

These periods vary from one year to another, likewise the intensity of the increases, but the general pattern remains the same, as demonstrated by the case of the dam Champagne (Renon stream), monitored in 2009 and 2015 (Fig. 2).





Furthermore, the average temperature downstream of the structure was systematically higher or equivalent than that measured upstream.

Different types of time series were observed regarding the difference between upstream and downstream temperatures:

The most frequent (7/13) is the type observed on the dam of Champagne (Renon stream) in 2009 and 2015; the minimum daily temperatures (T min) are, most usually, higher downstream of the structure, but the maximum daily temperatures (T max) remain within the same magnitudes (Fig. 2, only one example is presented here).

In the other cases (6/13), both the minimum and maximum daily temperatures are higher downstream of the structure, which results in a homothetic lag between the two temperature time series (Fig. 3).

### 3.2    Magnitude of upstream/downstream differences

The two dominant patterns can be illustrated by plotting the minimum and maximum temperature values at the site "Dompierre 2010" with a difference of order of + 1.5°C between the upstream and downstream of the site, comparing to "Neuf  2016", where these values are the same for minimum daily temperatures, or even slightly negative for the maximum temperatures (Fig. 4).

### 3.3    Reduction in the daily amplitude of downstream temperatures compared to upstream temperatures

Daily amplitude of the downstream temperature compared to that of the upstream is reduced in 61.5% of cases. This difference averages 0.46% for the 13 cases. This reduction in amplitude is due to a daily minimum downstream temperature that is 0.96°C higher than that of the upstream.

### 3.4    Seasonal pattern

During the summer season, the differences in the daily mean temperatures upstream / downstream, are close or staggered during all the season. It is notable that the variability of the summer air temperature is much higher (range 17°C) than stream temperature (range 7.5°C) for these examples (Fig. 5), and that the daily water temperature is not well correlated to air temperature.

### 3.5    Site typology based on summer thermal regime

The median values of the daily temperature variables calculated over summer (from 01/07 to 01/09) permit distinguishing two major types of response to the presence of a small dam (Table 3).

A first group (A) is characterised by:

- a median of the differences upstream/downstream of the maximum daily temperatures lower than 0.5°C;
- a median of the differences upstream/downstream of the minimum daily temperatures between + 0.4 and 1.3°C;
- a median of the differences in daily amplitudes lower than - 0.2°C.

A second group (B) is characterised by:

- a median of the differences upstream/downstream of the maximum daily temperatures higher than 0.5°C;
- medians of the differences upstream/downstream of the maximum and minimum daily temperatures in the same order of amplitude.

In addition two subgroups can be distinguished: subgroup (B2) with medians of upstream/downstream differences of daily maximum and minimum temperatures higher than 1°C, i.e. net warming between upstream and downstream, and subgroup (B1) with values ranging from 0.3 – 0.8°C.





The distribution of the differences between the minimum and maximum temperature values during summer (fig. 6) confirms the difference between these two groups, with greater variability for group B.

### 3.6    PCA Results

The first axis of the PCA analysis (78.3 %) is correlated to all temperature daily variables (calculated as differences between downstream versus upstream), in particular to the maximum daily temperature difference (Tmax_diff). The second axis discriminates the daily amplitude difference (Range_diff) with the minimum temperature (Tmin_diff) difference (Fig. 7).

For the determinants, the water residence time is the most correlated variable to the first axis F1, the size of the reservoir (surface, volume, length) correlates to both the first and second axis. The other physical-geographical characteristics related to the size of the watercourse (watershed, distance to the source), are correlated with the daily maximum temperature and associated with the second axis F2 (20.7 %); dam height has a very weak correlation with the axis F1.

The projection of the site series on these axes shows a strong spreading along the first axis. The dams measured two different years stay within the same range on this axis (Fretaz and Champagne) (Fig. 8).

Groupe B1 and B2 are distinguished by respectively the first and second axis association. This can be linked to the determinants of strong residence time influence for group B2, whereas group B1 is mainly characterized by the size of the impoundment (large impoundments, yet with relatively smaller residence time and thus less exacerbated thermal regime effects).

### 3.7    Focus on temperature pattern in short period of time

Looking more specifically on a short period of time (three consecutive days), differences in the diurnal variation of the temperature of the river upstream and downstream of the dam shows that for the first group A, the maximum water temperatures upstream and downstream are close, while the minimum temperature downstream does not return to that of upstream (Fig. 9A). In the second group B the water temperature difference between upstream and downstream are more important and remain persistent during all the day period (Fig. 9B).

### 3.8    Upstream/downstream temperature regime differences in terms of number of days exceeding a threshold

The warming of the water - in particular its ecological relevance - can be illustrated by repeatedly exceeding a temperature threshold during the summer season. For example, for the maximum daily temperature threshold of 22°C (arbitrary value), one can compare the effect of warming by the proportion of number of days exceeding this threshold upstream and downstream of the dam (Fig. 10).

This shows that some structure have a distinct difference between upstream and downstream.

### 4    Discussion

The number of small dams in streams is known to outweigh by several orders of magnitude that of large dams (> 15 m) (Downing et al., 2006; Poff and Hart, 2002; Verpoorter et al., 2014). Paradoxically, their effect on the thermal regime is much less well known and little documented as such (Downing, 2010; Ecke et al., 2017; Smith et al., 2017). This is challenging to identify and generalize the significant drivers of a realistic thermalscape (Isaak, 2017), an essential knowledge to understand the current ecological status of rivers and to predict with sufficient realism future changes with different





scenarios of climate change. In addition, the summer period, with the highest temperatures, appears to be potentially the most impacting for aquatic organisms and as such requires special attention (Kemp, 2012; Zaidel, 2018).

The purpose of this study was to quantify the downstream impacts of different types of small dams on summer water temperature in lowland streams. We investigated these effects in 5 lowland streams in the Bresse region (11 dams) in different climate years (12 summer time-series in warmer and drier years than normal and only one series in a colder and wetter year 2014 (Table 2). We observed that the dams altered the downstream thermal regime at sites, in 23 % of the time-series with a > 1°C elevation of the maximum daily temperature, and for 77 % in the range between -1°C and +1°C; the mean increase of the minimum daily temperature was 1°C, with 85 % of the time-series > 0.5 °C. This increase can reach 2.4°C at certain structures (Dompierre, Fig. 8).

Our results corroborate the reviews and meta-analyses in the general trend for a warming effect being generally within a range of 0 to 3°C (Lessard and Hayes, 2003; Maxted et al., 2005; MBaka and Mwaniki, 2015; Ecke, 2017, 27 studies; Means, 2018, 24 sites; Zaidel, 2018, 18 sites). Occasionally, higher values are found to reach as much as 7°C (Margolis et al., 2001; Carlisle et al., 2014), 3 to 6 °C (Fraley, 1987; Lessard and Hayes, 2003, for a part of their sample; Dripps and Granger, 2013). One possible explanation is that such sites correspond to very large impoundments in comparison with low natural flows, with large areas exposed to solar radiation, which pleads for an analysis of the physical characteristics of their structures. There are also situations where the downstream temperature is lower than the upstream temperature, that we only observed in one situation in our study (Moulin Neuf, Reyssouze, Fig. 8). These values are considered by the authors, particularly in the case of certain beaver dams, as particular configurations where the existence of a structure modifies the equilibrium conditions of the alluvial groundwater table, which under increased pressure can supply the downstream end of the structure with cooler water (Majerova et al., 2015; Weber et al., 2017). The morphology of the structures therefore appears to be fundamental; impoundments with high-head dams and a small surface area would have cooler downstream temperatures, whereas impoundments with low-head dams and a large surface area would have warmer downstream temperatures (Fuller and Peckarsky, 2011, Rocky Mountains in Colorado; Means 2018, Upper Columbia River). Concerning the possible effect of the alluvial groundwater, we wanted to avoid it as much as possible by choosing the downstream stations as close as possible to the dam (< 100 m). Despite the operating precautions taken, it is possible that the site Moulin Neuf, Reyssouze, which had a particular morphology with secondary channels, was still influenced by groundwater inflows.

The other major effect is the reduction in diurnal thermal amplitudes downstream of the smallest dams. We observed that the daily amplitude of the downstream temperature compared to that of the upstream is reduced in 61.5% of studied cases, in the same proportion than the observations of Zaidel (2018) for 58 % of the 30 structures in Massachusetts. Kemp et al. (2012) concluded also that the main influence of beaver ponds was a reduction in river temperature fluctuations. Amplitude reduction is due to a daily minimum downstream temperature that is 0.96°C higher than that of the upstream. Studying 24 beaver ponds in Washington State, Means (2018) observed also that the minimum temperature downstream was 0.8°C higher compared to minimum temperature upstream.

## 4.1 What physical variables are important?

The recording of temperatures alone and the identification of warming effects is not sufficient on its own to understand and generalize situations. The general laws of thermodynamics governing general heat balances must be carefully applied, taking into account the extreme diversity of physical configurations of dams that modify the state of radiation and the residence time of the flow.





To date, the effect of small dams on temperature has been little studied, either in isolated case studies (one stream, Kornis et al., 2015; Majerova et al. 2015; Smith et al. 2017; Weber et al. 2017), often as a secondary variable and with incomplete information about the characteristics of dams (Kemp et al., 2012).

Nevertheless, as early as the pioneering studies (Cook, 1940), candidate variables emerged concerning the exposed surface
subjected to radiation and the residence time of the water. Poff and Hart (2002) recalled that knowledge of the dam's water level or dam's height alone was insufficient to predict a thermal effect downstream. Caissie (2006) classified the environmental factors that influence the thermal regime of streams into four groups: (i) topography; (ii) streambed; (iii) stream discharge (friction, volume of water, slope, turbulence, inflow/outflow); (iv) atmospheric conditions (solar radiation, air temperature, wind speed/humidity, precipitation, evaporation/condensation). We have described precisely different
candidate metrics for the observed sites (Table 1). Our results show that the sites can be grouped in three major types presenting different sets of changes for maximum daily temperature (T max), minimum daily temperature (T min) and daily temperature amplitude.

Two distinct behaviours were observed in the 13 time series (Fig. 8): (i) group A characterised by an impoundment effect that reduced the amplitude of the daily temperature and increased the minimum temperature. This can be attributed to lower
nocturnal cooling of the mass of water in the impoundment – the median of the differences between the maximum temperature upstream and downstream of the structure was limited to 0.3°C at most; (ii) groups B1 and B2 characterised by warming with an increase of daily minimum and maximum temperatures with a change in amplitude.

In this dataset, the two distinct behaviours can be explained by two parameters: residence time (daily flow/impoundment volume) and the surface area of the impoundment. Group A is characterised by a residence time less than or equal to 0.7 days
and an impoundment surface area smaller than 35 500 m². The second group is characterised either by a long residence time, Dompierre dam (8.4 days) with an average surface area (10 900 m²), or by a surface area larger than or equal to 35 000 m², with a shorter residence time (0.2 days). In the first case, long residence time no longer allows the system to cool itself; the nocturnal input becomes negligible in the general heat exchange balance, whereas in the second case, heating linked to solar radiation (large impoundment surface area) becomes predominant in the general thermal balance.

To sum up, the first group is characterised by an impoundment effect on the increase of the minimum temperature, and on the reduction of the daily thermal amplitude without a significant change in the maximum temperature difference which fluctuates between - 0.6°C and + 0.3°C. The second group shows an effect on the increase of the minimum and maximum temperatures, with the daily amplitude remaining between - 0.3°C and + 0.4°C. For the types of group B, the change in thermal regime is much clearer, with differences in the order of 0.6 to 2.4°C in the median value for summer between
upstream and downstream of the structure, indicating a clear break in the temperature between the upstream and downstream ecosystems, notably during very hot periods. A larger sample of dams of B group type would permit a more quantitative characterisation of the dams (surface area, residence time, morphometry of the impoundment) above which the change of thermal behaviour influenced by an impoundment seams to appear. One potential path for deepening research is regionalisation as a function of thermal regimes and their governing factors (characteristics of aquifers/climate/bed material
conductivity).

In summary, mean air temperature (Fig. 5) or dam height (Fig. 6) are poor predictors of daily temperature fluctuations and their regime in summer. On the other hand, the median residence time and the surface area subjected to radiation are the main variables that explain the differences in thermal regime induced by small run-of-river dams. These variables are candidate to





generalize results. Although, this generalization necessities more precise information than the actual dam description in the vast majority of available public databases.

## 4.2 Elements of analysis of the thermal regime from an ecological perspective

Similar to what Poff et al. (1997) had proposed to analyse hydrological regimes in terms of intensity, duration, frequency, seasonality and rhythm of change, Bevelhimer and Bennet (2000) proposed to evaluate ecological impact with an approach

that takes into account the duration and amplitude of exposure to high temperatures and the recovery from stress during periods of lower temperature.

Considering a moderate absolute temperature increase between 0°C and 1°C downstream small dams, the majority of sites belonging to groups A and B1 present a low risk with regard to the potential change in fish communities. On the other hand, based on the values of 2°C of Hay et al. (2006) or of 3°C in the biotypology of Verneaux (1977), the higher values of our

group B2 comprised between 1.2 and 2.4 °C (Fig. 8) are likely to influence the composition of fish communities, especially regarding the conservation of certain species close to the threshold of their thermal comfort. It can also contribute to accentuate the general metabolism in the stream, possibly leading to the unwanted proliferation of algae, a less stable oxygen cycle, and stronger effects of toxic compounds (Heugens et al., 2001 in Souchon and Tissot, 2012).

We have chosen temperature > 22°C as an illustrative threshold known to be a thermal stress benchmark value for salmonids

especially for brown trout, *Salmo trutta* (Elliott and Elliot, 2010: upper critical incipient lethal temperature for alevins considered as a very sensitive stage; Ojanguren et al., 2001: general activity of brown trout juvenile). We also know that thermal regime and threshold values are important for the life cycle of aquatic invertebrates (Ward, 1976; Brittain and Salveit, 1989), and it is possible that changes in natural temperature regimes may be as important as altered stream flows to the ecological impacts of dam operations (Olden and Naiman, 2010).

The figure 4 shows two examples of stream thermal regime: site Neuf is clearly naturally not favourable for this species, even if the amplitude of downstream/upstream variation of temperature is weak; at the opposite, there is a clear non favourable regime at site Dompierre induced by the dam, numerous values of daily maximum temperature being above 22 °C until 27 °C.

On the scale of several days, it is also important not to underestimate the influence of exposure cumulated over time to

temperatures close to the maximum tolerable temperatures (Tissot & Souchon, 2010) for which the incidence of temperature variations has an impact on biological communities, as shown by Lessard and Hayes (2003) in 9 streams in Michigan (USA). For this purpose, we have chosen to represent the percentage of summer days with mean daily temperature > 22°C, upstream and downstream dams (Fig. 10). The majority of sites are naturally not favourable in summer for this species. But for more favourable sites on the left of the figure, Dompierre or Thuets, there is a clear shift towards an elevated percentage of number

of days > 22 °C, from less than 20 % upstream to more than 40 % downstream.

On the daily scale, it is necessary to not only consider the maximum tolerable temperature, but also its duration of influence, since the temperature of nocturnal remission and its duration must be sufficient for organisms to repair their heat stress proteins. Schrank et al. (2003) and Johnstone and Rahel (2003) suggested that daily minima provide a respite from elevated daily maximum temperatures if there is sufficient time to repair protein damage (McCullough et al., 2009). We examine 2

examples of daily temperature regime during 3 days in August at sites Caillou (type A) and Revel (type B2) (Fig. 9). At Caillou (Fig. 9A), the nictemeral natural variation offers remission temperature for brown trout, with several hours at temperature < 20 °C each day; the situation is less favourable downstream with no sufficient time below this temperature. At





Revel (Fig. 9B), the observed thermal daily pattern is analog, but the structure of type B2 exacerbated the warming of water accompanied by less remission periods.

Without appropriate biological data, it is difficult to know how minimum and maximum water temperatures affect acclimation, performance, and stress (McCullough et al., 2009). Exploring this question may be especially relevant because small dams impact stream temperature downstream, in a way which could be generalized by global warming in the future: raising daily minimum temperatures more than daily maximum temperatures, with a corresponding decrease in the diurnal temperature range and an increase in mean daily temperature (Easterling et al., 1997; Vose et al., 2005).

**4.3  Diversity of situations**

We measured variable warming effects according to the great diversity of situations present within a relatively modest geographical area (2025 km²), subjected to the same climate overall. We recalled the very high density of structures existing in this region (0.64/km or 1/1.56 km). The thermal landscape is therefore potentially very fragmented due to this fact alone. To have a realistic thermal landscape with which ecological dynamics are confronted in the long term, it would also be

necessary to take into account the linear features of rivers that are not protected by the shading of a riparian river (warming effect due to unfiltered radiation) or that are subjected to a supply of groundwater (cooling effect due to a cooler groundwater temperature than that of the river).

For periods of higher biological activity, such as the summer chosen in this work, we show that the correlations between air temperature and water temperature on a daily scale are not precise and that they must be mobilized with caution (Fig. 5). Our

work provides spatial generalization elements to better document the present and future thermal landscape. It is essential to help the authorities in charge of applying the environmental protection texts in force to prioritize the situations to be protected or restored by promoting greater thermal resilience over a significant portion of the length of a river.

Given the complexity and high variability of the river systems encountered in this study for river 3 to 5 Strahler orders, it seems essential to us, as Isaak et al. (2017, 2018), Steel et al. (2017), Dzara et al. (2018), to continue to conduct and densify

well targeted temperature monitoring before being able to model this variable with sufficient spatial and temporal resolution. This is a major challenge, insofar as it is the aquatic spaces that will be the site of major changes in the thermal and hydrological regimes to come with climate change, where tipping points in biotic distributions are likely to occur.

**5  Conclusion**

The impact of small dams on the temperature of streams has been poorly described in the scientific literature, and no clear

trends or determinants had been identified. More thorough analysis of time series of summer temperatures at small dams in a study area affected by probable warming according to the IPCC scenarios of global change (Val de Saone) allowed identifying the orders of magnitude of the contemporary impact according to the dimensions of these structures. The identified drivers of the temperature regime responses are the residence time and the impoundment surface submitted to radiation. These determinants are candidate to generalize results, but this would necessitate the gathering of more precise

information than the current dam descriptors in public databases.

**6  Acknowledgements**

We thank the local river management body, the Syndicat Mixte Veyle Vivante and its employees Laurent Charbonnier and Stéphane Kihl, for installing the measurement network, their help for field monitoring and their valuable practical advice. We




also thank the regional branch of the Ministry of the Environment (Dreal Rhône-Alpes; formerly DIREN SEMA) for
punctual gauging data, edition of 15 April 2002. The Rhone Mediterranean Corsica Water Agency provided financial support
which allowed recording the times series and analysing the data.

Earlier versions of the manuscript were improved by comments from… and … anonymous reviewers.

The authors declare no competing interests.

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





**Table 1. Physical characteristics of dams of the river and impoundments.**

| Stream name | Dam name | Watershed (km²) | Distance to the source (m) | Strahler order | Dam height (m) | Length (impoundment) (m) | Surface (m²) | Volume (m³) | Residence time (days) | Year of sampling |
|---|---|---|---|---|---|---|---|---|---|---|
| Veyle | Dompierre | 32 | 11167 | 3 | 1.2 | 500 | 10900 | 10500 | 8.4 | 2010 |
| Veyle | Fretaz | 78 | 22859 | 4 | 1.5 | 535 | 3500 | 2600 | 0.1 / 0.1 | 2014 / 2016 |
| Veyle | Montfalconnet | 125 | 38146 | 4 | 2.4 | 1200 | 14400 | 20160 | 0.5 | 2015 |
| Veyle | Peroux | 500 | 50886 | 5 | 2.4 | 2150 | 39200 | 53000 | 0.6 | 2015 |
| Veyle | Thuets | 350 | 43912 | 5 | 1.9 | 2950 | 57000 | 51000 | 0.6 | 2016 |
| Veyle | Thurignat | 640 | 60537 | 5 | 1.4 | 1500 | 34600 | 31165 | 0.2 | 2016 |
| Vieux Jonc | Cailloux | 67 | 11680 | 3 | 1.0 | 280 | 2340 | 1200 | 0.7 | 2009 |
| Renon | Champagne | 122 | 42368 | 3 | 1.5 | 405 | 2840 | 2130 | 0.7 / 0.5 | 2009 / 2015 |
| Reyssouze | Moulin Neuf | 209 | 48217 | 3 | 1.0 | 1800 | 35520 | 12420 | 0.3 | 2016 |
| Reyssouze | Peloux | 145 | 34842 | 3 | 1.5 | 1700 | 49930 | 17340 | 0.5 | 2016 |
| Solnan | Revel | 88 | 15431 | 3 | 1.8 | 3200 | 31140 | 28370 | 2.6 | 2016 |

**Table 2. Climatic characteristics during years of stream temperature monitoring (2009-2016).**


| Year (July – August) | Air temperature (difference from normal (°C)) | Precipitation (difference from normal (%)) |
|---|---|---|
| 2009 | +1.1 | 70 |
| 2010 | +0.3 | 50 |
| **2014** | **-1.8** | **165** |
| 2015 | +2 | 50 |
| 2016 | +0.3 | 70 |

(Source: https://www.infoclimat.fr station Lyon Bron normal (1991 – 2015))



**Table 3. Median values of differences between daily maximum (maxT diff.) and minimum temperatures (minT diff.) and the diurnal ranges (ranges diff.) between upstream and downstream of the run-of-the-river dams. Daily maximum upstream temperature (maxT upstream) is indicated to show the limited influence of the initial temperature on upstream / downstream differences.**


| Group | Run-of-the river dam (stream) | maxT diff. °C | minT diff. °C | range_diff. °C | maxT upstream °C |
|---|---|---|---|---|---|
| A | Moulin Neuf (Reyssouze) 2016 | -0.6 | 0.5 | -1.0 | 24.0 |
| | Cailloux  (Vieux Jonc) 2009 | -0.4 | 0.9 | -1.3 | 18.1 |
| | Fretaz  (Veyle) 2014 | 0.3 | 0.7 | -0.3 | 19.4 |
| | Fretaz  (Veyle) 2016 | -0.3 | 1.2 | -1.4 | 21.2 |
| | Champagne (Renon) 2015 | 0.1 | 0.9 | -0.9 | 20.2 |
| | Montfalconnet (Veyle) 2015 | -0.1 | 1.0 | -0.8 | 19.8 |
| | Champagne (Renon) 2009 | -0.1 | 0.7 | -1.0 | 19.3 |
| B1 | Thurignat (Veyle) 2016 | 0.6 | 0.3 | 0.4 | 23.2 |
| | Thuets (Veyle) 2016 | 0.7 | 0.8 | 0.0 | 21.0 |
| | Peloux (Reyssouze) 2016 | 0.8 | 0.5 | 0.1 | 23.9 |
| B2 | Peroux (Veyle)  2015 | 1.1 | 1.1 | -0.3 | 21.3 |
| | Revel (Solnan) 2016 | 2.1 | 1.7 | 0.1 | 21.9 |
| | Dompierre (Veyle)  2010 | 2.4 | 2.2 | 0.4 | 18.2 |





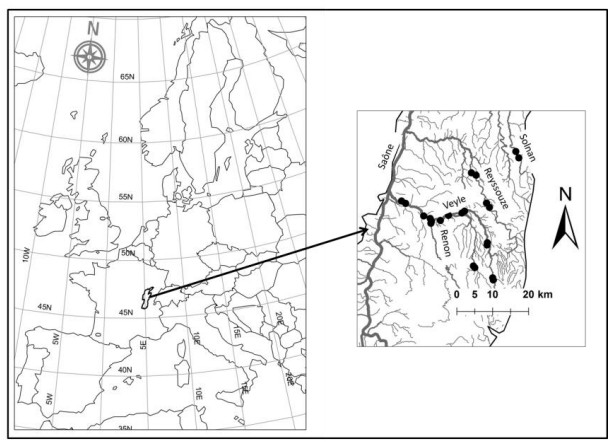

**Figure 1. Location of the study area, the Bresse Region – The black points on the right map indicate temperature recording sites.**

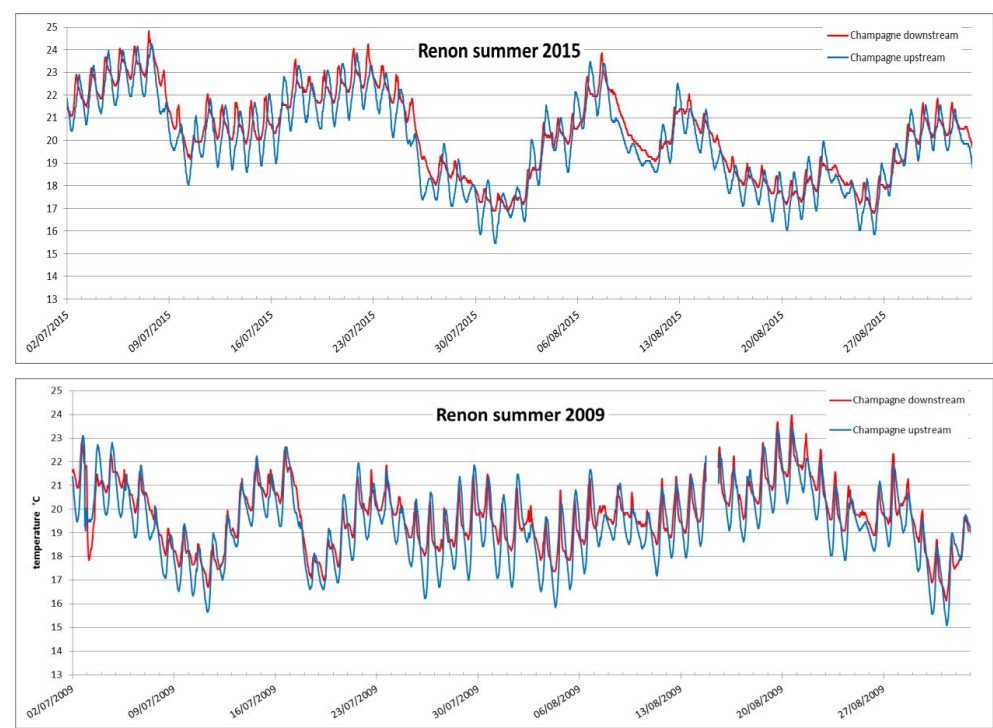

**Figure 2. Time series of water temperature upstream (blue) and downstream (red) of the dam Champagne, Renon stream respectively in years 2015 and 2009.**






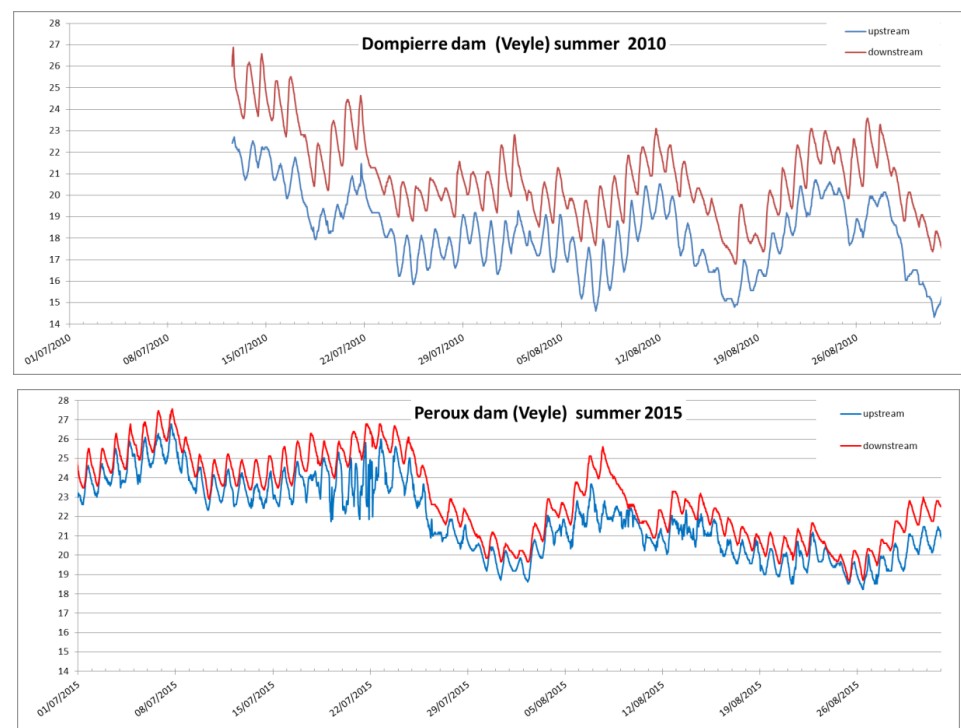

**Figure 3. Time series of water temperatures upstream (blue line) and downstream (red line) of the dams of Dompierre and Peroux, Veyle stream (2010 and 2015, two warm summer years, respectively + 1.1 °C and 2° C, Table 2).**

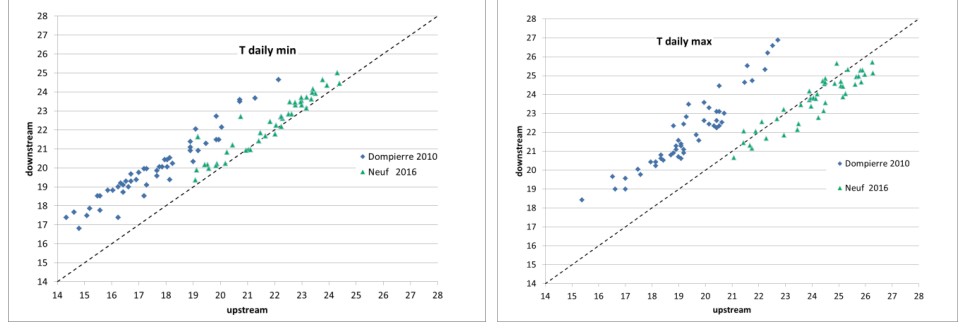


**Figure 4. Minimum and maximum daily temperatures upstream and downstream of the dams-of-the river (Neuf site, Reyssouze stream in 2016; Dompierre site, Veyle stream in 2010).**





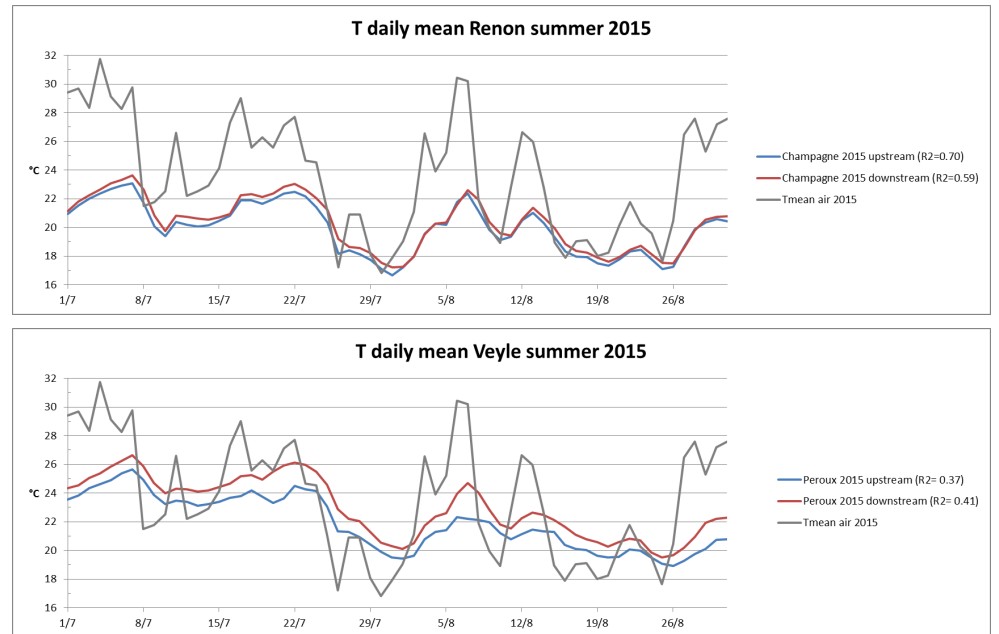

**Figure 5. Mean daily temperature upstream (blue), downstream (red), air (grey) at sites Champagne (Renon) and Peroux (Veyle). Year 2015.**



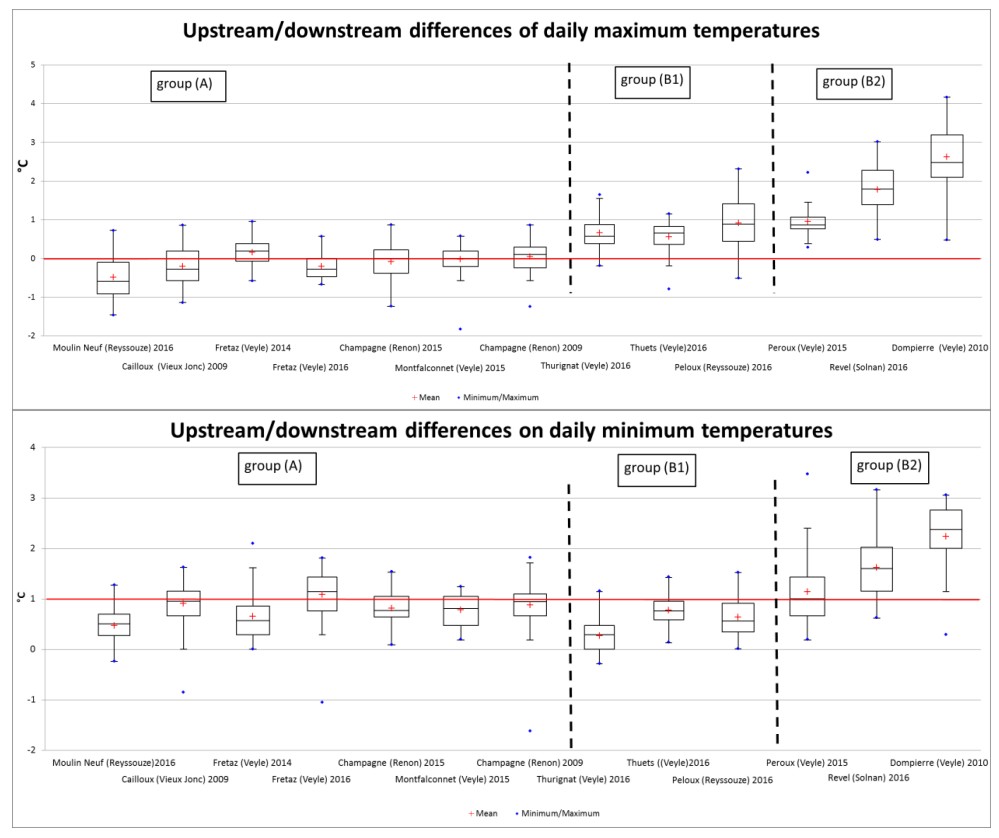

**Figure 6. Box-plot distribution (25% - 75 %) of upstream/downstream differences of daily maximum and minimum temperatures for all the time series studied. (Red lines: 0°C for daily maximum temperature and 1°C for daily minimum temperature are drawn to help reading).**






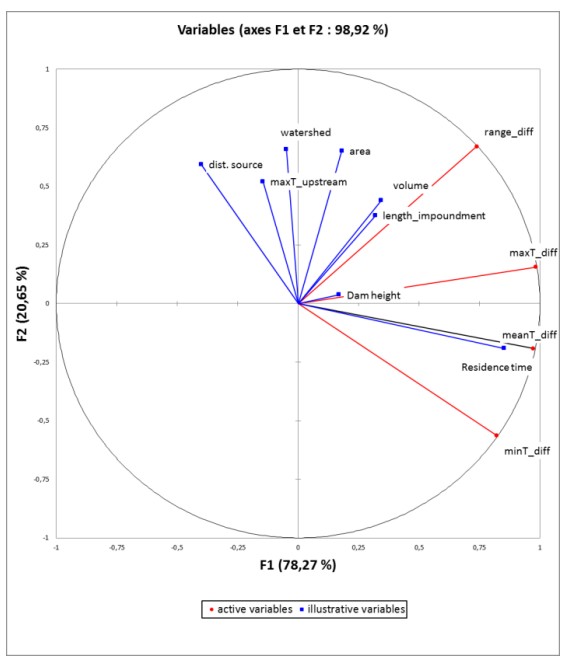

**Figure 7. PCA analysis. Correlation circle with temperature as active variables, and physical characteristics as illustrative variables.**

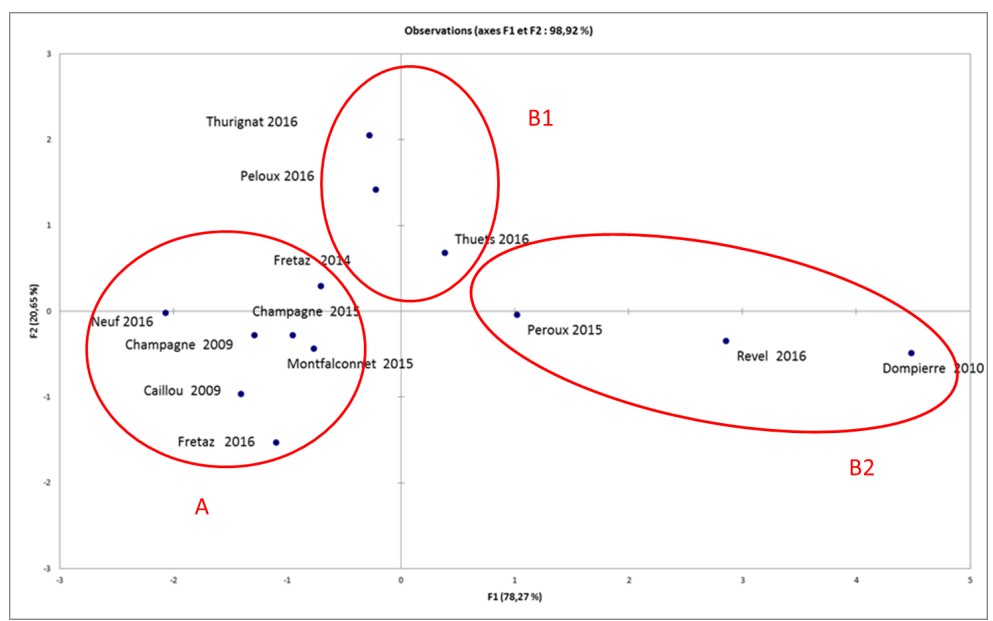


**Figure 8. PCA analysis. Scatterplot of sites * years. Ellipses are drawn to visualize the identified summer thermal regime types.**



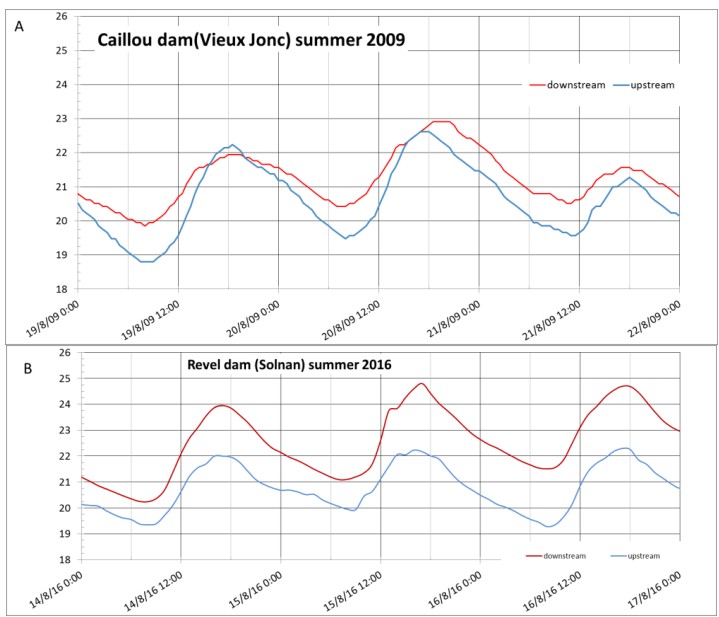

**Figure 9. Time series of water temperatures upstream (blue line) and downstream (red line) of the dams of A/ Caillou (Vieux Jonc stream) and B/ Revel (Solnan stream) focused on three days during August.**

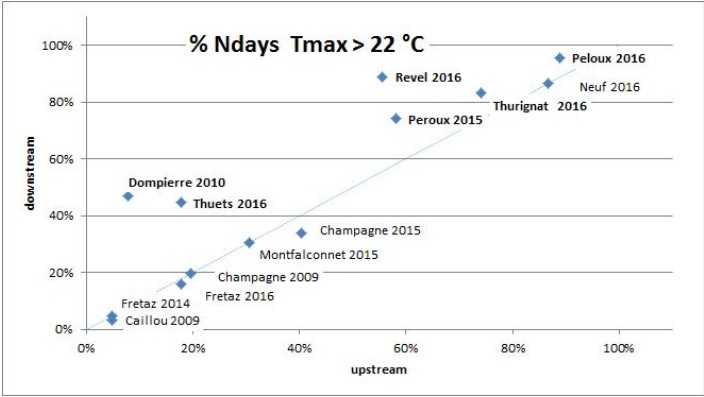

**Figure 10. Percentage of number of summer days with a diurnal maximum temperature of water greater than 22 °C, upstream and downstream each site monitored in the study.**