# Peer review of "Determinants of thermal regime influence of small dams"

_Hydrology and Earth System Sciences, 2019_

## Referee Comment (RC1) · Anonymous Referee #1 · 2 May 2019

General Comments: The topic of this paper is of high importance, and the research is critically needed as the authors correctly point out that our knowledge base of the impact of small dams on water temperature (and stream ecosystem function in general) is woefully lacking. The data set the authors have collected appears robust, and with a sufficient number of sites to make a useful contribution to the literature. I find the analysis of the data to severely lacking, and the presentation of the results to be mainly using individual sites as examples that are difficult to judge if they are representative. The discussion is quite thorough and insightful, but without adequate data analysis, I felt that the conclusions were not well supported.

Specific Comments: 1. Figure 2 – why present years in reverse chronological order? Also, why this stream and these years? If possible, it would be preferable to compare

2014 (cold wet year) with 2015 (warmest, dry year in data set). 2. General – figures don't do a very good job of illustrating points made in text in results. I question whether all the figures are needed (e.g., Figure 3). Figure 5 – presenting time series does not show correlation between two variables –one would need to plot air temp vs. water temp to show directly. Figure 4 – never covered in results section. 3. The authors mention differences in mean temperature, but never provide this information in a table. Further, they report median differences without justifying why this metric instead of means. I feel medians can be a useful indicator of central tendency, but the mean is also useful, and needs to be presented if it is discussed. 4. Section 3.4 – authors state that air and water temperatures do not correlate, but did not perform a correlation analysis. 5. Section 3.5 – how were these groups distinguished (meaning, what formal method was used). My impression is that the investigators did this "by eye", which is not acceptable in my view. A formal cluster analysis would be much more appropriate. Moreover, I think it is hard to defend splitting out groups with such a small number of sites. 6. Section 3.6 – in the methods, the authors state that they used mean temperatures in the PCA analysis, but this doesn't show up in the results. Further, the reporting of the PCA results is very incomplete. Loadings of the various variables is needed, as is some criterion for determining what are the significant correlations. I can't say I understand fully how to interpret the circle correlation plot. 7. Section 3.7 – this section does not provide a synthetic view of any of the data, and the intent of this section is unclear. Suggest removing it entirely. 8. Section 3.8 – the arbitrary nature of this analysis provides little insight or direct ecological interpretation. In the discussion the authors correctly indicate that the choice of a 22 degree is actually not arbitrary, but has a basis in that temperatures above this point are generally deleterious to salmonids. Although I think this section could be a valuable contribution by the research, the fragmented presentation leads me to suggest removing it entirely. 9. In the discussion, the authors talk about different years (hot vs. cool, or wet vs. dry), but none of the analysis really looks into this. I think it is an important point, so would like the authors to explore and quantify this in a reasonable way. 10. In the introduction and

discussion, the authors talk about the importance of dam and reservoir size, but don't do any formal analysis. At a basic level, it would seem that correlation or regression of reservoir area, and another analysis with residence time, on the response variables of mean temperature difference, mean difference in maximum temperature, and mean difference in minimum temperature would be an important starting point. 11. The discussion of biological effects was quite thorough.

Technical Comments: 1. Many grammatical errors – far more than is appropriate for a scientific reviewer to make edits on, but these need to be addressed before publication. 2. The citation for Dunham et al. is incomplete, but I applaud investigators for addressing instrument calibration issues, which are often ignored!

---

## Referee Comment (RC2) · Anonymous Referee #2 · 11 May 2019

General comments:

In general, the paper discusses a relevant research issue, as is discussed based on the literature in the discussion. It is apparently based on an interesting dataset (though with some limitations, mentioned below), but the presentation and discussion of the results is relatively poor and not very clear, and calls for major revisions.

It should be made more clear (in the introduction etc.), that the results are probably not easily transferrable to other areas, as the choses study sites are quite homogenous (focus on a certain region of France). Furthermore, the study would greatly benefit from including more temperature data from the same site for several years – one would expect to also see quite some inter-annual differences. As this does not seem to be possible, the authors should at least discuss this shortcoming. Especially as the

authors try to hint at a regionalization (e.g. at the end of section 4.1), this should be discussed better: What, for example, about the different groundwater regimes – are we talking about gaining or losing rivers? Etc.

The overall result – that the most important drivers of temperature regime changes in dams are residence time and surface area are not particularly surprising. Discuss this. (maybe one could even come up with some empirical linear relationship or empirical model, including those parameters, and water temperature, air temperature, solar radiation etc.?)

Specific comments:

Section 1: Please include some more general explanation on why the whole issue of dams changing the thermal regime is relevant (make your motivation more clear)

Line 27: "These determinants are candidate to generalize results" – sentence a bit unclear, please reformulate

Line 47: "During summer, the factors leading to warming are: (i) the input of heat from upstream" – maybe you should be a bit more specific here. Mention why you focus on summers. What do you mean by the input of heat from upstream? Tributaries that are warmer than the main stream?

Line 50: If you talk about different anthropogenic influences on stream temperature, you probably also should mention cooling water from power plants etc.

Line 56: > 15 m of what?

Line 61 ff: These two "predictions" you are mentioning from 1983 and 1990 should be verified by now? Can you say something about this?

Line 84: With a height smaller than 5m?

Line 88ff: Be more precise here. There are few articles even considering temperature effects? Those are the 43 sites or articles?

Line 106: "with closed riparian canopy or aquifers" – what do you want to say here?

Line 106ff: "This is the reason why it seems preferable in a first study to focus on the single effects of the impoundment immediately downstream the dam." – please reformulate/make your motivation more clear. How exactly is this resulting from the above?

Line 130: How is a "day of heat wave" defined?

Section 2.2: Mention right away in the text how many dams you study. And how did you chose those specific sites?

Line 145: Make it clear that the temperature sampling was performed for single summers (or two) per site, between 2009 and 2016

Section 2.5: Please elaborate further on how you performed your PCA. Illustrative variables are explanatory variables? "In order to identify characterization of the impacts of the different dams" – reformulate, unclear!

Section 3.2/Fig. 4: I understand that the scatter plot for Dompierre shows "type 2", so like in Figure 3. However, Neuf in Fig. 4 does not show "type 1", like in Figure 2, because there is almost no difference between minimum temperatures up- and downstream. And, why don't you simply show the same data in your timeseries plots (Fig. 2 and 3) and the scatterplot (Fig. 4) to illustrate the two types. Also, better to combine the figures and make the two types more clear by that.

Section 3.3: 0.46% of what?

Section 3.5: Specify how you calculate your differences (downstream – upstream?). And don't groups B1 and B2 both exhibit net warming? Be more precise.

Section 3.7: Confusing to speak of "short period of time" or "three consecutive days" – what you actually do is to look at shifts in intra-daily temperature variation.

Section 4, first paragraph: Some of this would be better in the introduction. Same

applies to first two paragraphs of section 4.1.

Line 317, 318: Again, specify the sign of your temperature differences.

Line 344ff: Is Salmo trutta a common species in the rivers of your test sites?

Line 378: "The thermal landscape is therefore potentially very fragmented due to this fact alone." What do you mean by this and the following sentences?

Line 385: Please specify which "spatial generalization elements" you mean.

Technical comments:

Be consistent with thousand separators (for example, you have 2 710000, 96 222, 59071)

Be consistent on how to write "run-of-the-river dam".

Line 38: Why do you cite Rader et al., 2007 as part of the review by Ellis and Jones?

Line 42: "precipitation", not "precipitations", this comes up several times

Line 68: reformulate to "they are expected to increase downstream water temperature" or similar

Line 78: "(ROE, sept 2017)" why is this cited this way?

Line 59: "water temperature patterns for tens of km"?

Line 72ff: "very imprecise depending on national databases. For example, the International Commission on Large Dams"

Line 90ff: "Dripps et al. (2013)...." – please reformulate, sentence unclear

Line 95 ff: "Hayes et al. (2008) in the region of the Great Laurentian Lakes" – all this paragraph contains typos and grammar mistakes, please revise

Line 101: Maybe "explaining variables" is a better term

Sector 2.1: Please revise language. Remove repetitive "on a basis of 230 000 km streams with permanent flow"

---

## Referee Comment (RC3) · Anonymous Referee #3 · 14 May 2019

General comments:

The purpose of this study was to quantify the downstream impacts of different types of small dams on summer water temperature in lowland streams. The topic of this manuscript is of high importance, and the research is critically needed since water temperature could impact the structure of aquatic communities and the functioning of the aquatic ecosystem as stated by the authors. The data set on water temperature the authors have collected seems to be robust, and with quite enough number of sites. I personally appreciated the calibration process made for the instruments to insure reliable data. The discussion is quite thorough and insightful, but more focus on literature review (others work) rather than focusing on the discussion of the current work. I found that data analysis severely lacking, and the presentation of the results to be

using individual sites as examples that are difficult to judge if they are really representative. Therefore, without adequate data analysis I felt that the conclusions were not well supported. The language used is not sufficiently comprehensible and needs to be improved before publication. Many other specific and technical comments can be found below.

Specific comments: (P=Page, L=Line)

1. P5, L159: Why authors calculate median differences and not mean? Please justifying why this metric instead of means.

2. Section 3.5: What is the scientific method used for group clustering?

3. Section 3.7: the results presented in this section are unclear and the purpose of presenting such results is unclear as well. I found it very hard to link this section with the discussion section. This would be easy for the reader if the results and discussion section were compiled in one section.

4. P7, section 3.8: Authors mention that the maximum daily temperature threshold of 22°C is arbitrary value. While later in the discussion, the authors indicate that the choice of a 22°C is actually not arbitrary. I suggest that authors delete the word arbitrary and explain the basis of this threshold choice.

5. P8, L255: the authors mention warmer, drier, colder and wetter years. Please discuss how these classifications are made?

6. P18: Fig.4: what is the reason for comparing temperature of different sites (Dompierre and Neuf) in different years (e.g. 2010 and 2016).

7. P19: Fig.3 caption: the authors state "Time series of water temperatures upstream (blue line) and downstream (red line) of the dams of Dompierre and Peroux, Veyle stream (2010 and 2015, two warm summer years, respectively + 1.1 °C and 2° C, Table 2)", but when looking back in table 2, I have seen that air temperature difference from normal in 2010 is very small (+ 0.3) and NOT +1.1. The +1.1 °C air temperature

difference from normal is in the year 2009. Therefore, 2009 is almost four times warmer than 2010, hence one may expect the comparison between 2009 and 2015 instead of 2010 and 2015?

8. P19: Fig.3: Since air temperature difference from normal in 2010 is very small (+ 0.3), why the difference between upstream and downstream water temperature at Dopmierre dam is very high? This cannot be due to long residence time and average surface are in absence of warm condition, so what could be the reason/s?

9. It is insecurely to compare 2014 (cold and wet year) with 2015 (warm and dry year) for at least one site (e.g. Dompierre dam) to see the effect of air temperature.

Technical corrections:

1. P18: in Fig.2 caption, what is the word "respectively" refer to?

2. P1, L18-19: "The mean increase of the minimum daily temperature was 1°C, with 85 % of the time-series showing an increase > 0.5 °C", this sentence is not clear or grammatically incorrect.

3. P2, L63-64: "surface release reservoirs", should read "surface reservoirs' release".

4. P5, L148-149: "in the main flow of the channel" should read "in the main flow channel".

5. P5, L151: "method Dunham et al. (2005)." should read "method introduced by Dunham et al. (2005)".

6. P5, L157: the authors state that "and the median values were recorded for the period", how do you record the median? It should read "calculated" instead.

7. P6, L182: "Furthermore, the average temperature downstream of the structure was systematically higher or equivalent than that measured upstream" should read "Furthermore, the average temperature downstream of the structure was systematically equivalent or higher than that measured upstream".

8. These are limited examples and the paper contains more. All grammatical errors should be fixed before publication.

---

## Author Comment (AC1) · 15 Jul 2019

Dear Editor and Referees,

Thank you for the quality of your proofreading and comments; they have greatly improved the manuscript. We also appreciate your interest in the subject matter, which we think is of critical importance to managers across France and the world who are dealing with issues of small dam removal and ecological integrity. We believe we have substantially addressed all of the outstanding comments and issues, and we look forward to your second review of the work. All of the referees remarked on the issue of data representativeness, so we will briefly discuss this issue here. Data scarcity (i.e., lack of data across years within sites) is a primary challenge for understanding thermal

effects of small dams, and it is one of the primary reasons that we used a compiled dataset with data from field operators, which we bolstered with our own sampling. We acknowledge that using these two data sources may make reading and understanding a little more difficult, but we believe it enriches the analysis by increasing the number of time series and across-year examples, (though we agree this dataset is probably still insufficient to draw broad conclusions). Hence, we are aware of the issues with the dataset, and we have added text throughout to underscore this issue. However, we feel that the analysis and general results are valid and useful, regardless of data scarcity issues, which every study must deal with. Throughout the manuscript, we have made major revisions based on the referees comments and suggestions. The major changes are: - use of new statistical analysis methods to strengthen the robustness of the results, - improved consistency between points raised in the comments and proposed figures, - grammatical quality review: a final revision of English was done by a native speaker.

General comments : " the presentation of the results to be mainly using individual sites as examples that are difficult to judge if they are representative."

Response: An improvement in the presentation and choice of sites selected as examples has been modified in the final text.

Specific comments: "1. Figure 2 – why present years in reverse chronological order? Also, why this stream and these years? If possible, it would be preferable to compare 2014 (cold wet year) with 2015 (warmest, dry year in data set)."

Response: The aim was to highlight that the same site presented the same "patterns" of summer time-series for different years, regardless of the climatic characteristics of the year. Based on this comment, we have changed this example (new Figure 2) to compare a cold and humid year (2014) with a normal and dry year (2016) at another site (Veyle stream, Fretaz site): the structure of the thermal patterns between upstream and downstream is preserved.

The new text L180 to 189 is modified as

Previous text: L180 to 189 These periods vary from one year to another, likewise the intensity of the increases, but the general pattern remains the same, as demonstrated by the case of the dam Champagne (Renon stream), monitored in 2009 and 2015 (Fig. 2). Furthermore, the average temperature downstream of the structure was systematically higher or equivalent than that measured upstream. Different types of time-series were observed regarding the difference between upstream and downstream temperatures: The most frequent (7/13) is the type observed on the dam of Champagne (Renon stream) in 2009 and 2015; the minimum 185 daily temperatures (T min) are, most usually, higher downstream of the structure, but the maximum daily temperatures (T max) remain within the same magnitudes (Fig. 2, only one example is presented here). In the other cases (6/13), both the minimum and maximum daily temperatures are higher downstream of the structure, which results in a homothetic lag between the two temperature time-series (Fig. 3).

Replaced by

These periods vary from one year to another, likewise the intensity of the temperature increases, but the general pattern remains the same, as demonstrated by the case of the dam Fretaz (Veyle stream), monitored in 2014 (a cold and humid year) and 2016 (a more normal year, Fig. 2; Table 2). We observed two consistent pattern in upstream/downstream thermal regimes. In the first pattern, the daily minimum temperature is higher downstream, but the daily maximum temperature stays relatively constant (Fig. 2). We note that these upstream/downstream differences were muted in 2014, the cold and humid year (Fig. 2). This thermal pattern (i.e., where the minimum temperature increases downstream, but not the maximum temperature) is observed in 7 out of 13 cases (Table 3). In the other cases (6 out of 13; Table 3), we observed a second pattern, where both the minimum and maximum daily temperatures are higher downstream of the structure, which results in a consistent shift between the two temperature time-series (Fig. 3, selected examples: Dompierre dam 2010 and Peroux

dam 2015, Veyle stream).

"2. General – figures don't do a very good job of illustrating points made in text in results. I question whether all the figures are needed (e.g., Figure 3). "

Response: Fixed; see above

"Figure 5 – presenting time-series does not show correlation between two variables –one would need to plot air temp vs. water temp to show directly. "

Response: We modify Figure 5 and the text as follows:

Previous text: L200 to 204 During the summer season, the differences in the daily mean temperatures upstream / downstream, are close or staggered during all the season. It is notable that the variability of the summer air temperature is much higher (range 17°C) than stream temperature (range 7.5°C) for these examples (Fig. 5), and that the daily water temperature is not well correlated to air temperature.

Replaced by

During the summer season, the upstream/downstream daily maximum water temperature differences are not well correlated with air temperature for the same periods. For example, a simple linear regression between daily maximum air temperature and daily maximum water temperature differences indicates that air temperature explains only 0.3% of the variability in upstream/downstream thermal regime shifts (Fig. 5).

"Figure 4 – never covered in results section."

Response: We previously covered figure 4 in section 3.2 but now we changed the text to better explain the observed pattern. We also changed the site "Neuf" to "Fretaz 2014".

Previous text: L 191 to 194 The two dominant patterns can be illustrated by plotting the minimum and maximum temperature values at the site "Dompierre 2010" with a difference of order of + 1.5°C between the upstream and downstream of the site, comparing

to "Neuf 2016", where these values are the same for minimum daily temperatures, or even slightly negative for the maximum temperatures (Fig. 4).

Replaced by

The two dominant patterns of temperature differences are further illustrated by plotting the minimum and maximum temperature values at the site. For example, at Dompierre in 2010, we observed a consistent shift of approximately +1.5°C (both maximum and minimum daily temperature) between the upstream and downstream of the dam (Fig. 4A). In contrast, at Fretaz in 2014, this shift is dampened, and temperature values between upstream and downstream follow a 1:1 relationship (Fig. 4B).

"3. The authors mention differences in mean temperature, but never provide this information in a table. Further, they report median differences without justifying why this metric instead of means. I feel medians can be a useful indicator of central tendency, but the mean is also useful, and needs to be presented if it is discussed."

Response: To avoid any confusion, we eliminate any reference to daily mean temperature. We also have modified the section 2.4 Data analysis to remove any confusion about using mean temperature (L 156 to 159).

Previous text: L156 to 159 To determine if the dams alter the temperature regime, the minimum, average and maximum temperatures and amplitudes were calculated for each full day recorded, and the median values were recorded for the period. The calculations of daily differences of maximum and minimum water temperatures were performed for each pair of upstream/downstream records, and the median of these differences over the recording period was calculated.

Replaced by

To characterize the influence of dams on stream thermal regimes we first calculated three variables: daily difference between upstream and downstream temperature 1) maximums, 2) minimums, and 3) ranges for each site and year. (..). With these data,

we then conducted the following analyses: 1. Median summer differences in maximum, minimum, and range between upstream and downstream (median is used instead of mean to characterize a season in order to limit the effect of a specific weather event), 2. …..

"Section 3.4 – authors state that air and water temperatures do not correlate, but did not perform a correlation analysis".

Response: Fixed with a new figure 5

"5. Section 3.5 – how were these groups distinguished (meaning, what formal method was used). My impression is that the investigators did this "by eye", which is not acceptable in my view. A formal cluster analysis would be much more appropriate. Moreover, I think it is hard to defend splitting out groups with such a small number of sites."

Response: The requested additional statistical analysis has been completed and we propose the following changes

We add description of the statistical method used Previous text: L 159 Finally, we propose a classification of the observed thermal behavior in 3 groups, based on differences between upstream and downstream dam daily maximum temperature, daily minimum temperature and daily amplitudes.

Replaced by 2.5 Site typology analysis We observed different thermal regimes in our data and wanted to classify them. To do so, we carried out a hierarchical cluster analysis using Euclidian dissimilarities matrix according to the Ward's method (1963) using daily dataset (n=807) of upstream/downstream differences between maximum and minimum temperatures obtained over all time-series. We forced the classification to integrate the different time-series effect by adding a complete disjunctive table differentiating each time-series to the data set. This procedure makes it possible to group the data first by time-series, then in a second step to differentiate them from each other (i.e., to differentiate site thermal regimes).

Previous text: L 204 to 217 3.5 Site typology based on summer thermal regime The median values of the daily temperature variables calculated over summer (from 01/07 to 01/09) permit distinguishing two major types of response to the presence of a small dam (Table 3). A first group (A) is characterised by: - a median of the differences upstream/downstream of the maximum daily temperatures lower than 0.5°C; - a median of the differences upstream/downstream of the minimum daily temperatures between + 0.4 and 1.3°C; - a median of the differences in daily amplitudes lower than - 0.2°C. A second group (B) is characterised by: - a median of the differences upstream/downstream of the maximum daily temperatures higher than 0.5°C; - medians of the differences upstream/downstream of the maximum and minimum daily temperatures in the same order of amplitude. In addition two subgroups can be distinguished: subgroup (B2) with medians of upstream/downstream differences of daily maximum and minimum temperatures higher than 1°C, i.e. net warming between upstream and downstream, and subgroup (B1) with values ranging from 0.3 – 0.8°C. Replaced by

3.5 Site typology The hierarchical cluster analysis applied on the values of the daily temperature variable differences over summer (from 1 July to 31 August) distinguished three groups: - a first group (A) characterized by: - a median of the differences upstream/downstream of the maximum daily temperatures less than 0.5°C; - a median of the differences upstream/downstream of the minimum daily temperatures between + 0.4–1.3°C; - a median of the differences in daily amplitudes less than -0.2°C. - a second group (B1) characterized by: - a median of the differences upstream/downstream of the maximum daily temperatures ranging from +0.6–1.2 °C; - a median of the differences upstream/downstream of the minimum daily temperatures between +0.3–1.1°C. - a third group (B2) is characterized by medians of upstream/downstream differences of daily maximum and minimum temperatures both higher than 1.2 °C (i.e., net warming between upstream and downstream)

Figure 6 changed.

"6. Section 3.6 – in the methods, the authors state that they used mean temperatures

in the PCA analysis, but this doesn't show up in the results. Further, the reporting of the PCA results is very incomplete. Loadings of the various variables is needed, as is some criterion for determining what are the significant correlations. I can't say I understand fully how to interpret the circle correlation plot."

Fixed; we have added new clarifying text.

Previous text: L 166 to 170 2.5 PCA analysis In order to identify the characterization of the impacts of the different dams, a principal component analysis (PCA) was carried out using the software XLStat (ADDINSOFT$^{TM}$) on the water temperature variables: downstream / upstream difference of the maximum, average and minimum daily temperature and daily temperature amplitude. The physical characteristics of the structures (Table 1) were used as illustrative variables to evaluate the correlations with the temperature variables

Replaced by

2.5 Ordination analysis To characterize the impacts of the different dams, a principal component analysis (PCA) was carried out using the software XLStat (ADDINSOFT$^{TM}$) on the three water temperature variables: downstream/upstream difference of the maximum and minimum daily temperature and daily temperature range. We used the median values for variables on each time-series in order to build an input matrix (13 occurrences for three variables). Then a complementary redundancy analysis (RDA) with automatic stepwise variable selection procedure was used to identify the physical dam characteristics (Table 1) that significantly explain the PCA results (ter Braak 1986). After the RDA identified the relevant physical dam characteristics, we conducted multiple linear regression between these characteristics and temperature variables to determine specific effect sizes of these characteristics on thermal regime. Ter Braak, C. J. F.: Canonical correspondence analysis: a new eigenvector technique for multivariate direct gradient analysis, Ecology, 67, 1167-1179, 1986.

Previous text: L 220 232 3.6 PCA results The first axis of the PCA analysis (78.3 %) is

correlated to all temperature daily variables (calculated as differences between downstream versus upstream), in particular to the maximum daily temperature difference (Tmax_diff). The second axis discriminates the daily amplitude difference (Range_diff) with the minimum temperature (Tmin_diff) difference (Fig. 7). For the determinants, the water residence time is the most correlated variable to the first axis F1, the size of the reservoir (surface, volume, length) correlates to both the first and second axis. The other physical-geographical characteristics related to the size of the watercourse (watershed, distance to the source), are correlated with the daily maximum temperature and associated with the second axis F2 (20.7 %); dam height has a very weak correlation with the axis F1. The projection of the site series on these axes shows a strong spreading along the first axis. The dams measured two different years stay within the same range on this axis (Fretaz and Champagne) (Fig. 8). Groups B1 and B2 are distinguished by respectively the first and second axis association. This can be linked to the determinants of strong residence time influence for group B2, whereas group B1 is mainly characterized by the size of the impoundment (large impoundments, yet with relatively smaller residence time and thus less exacerbated thermal regime effects).

Replaced by

3.6 Ordination results The first axis of the PCA analysis (74.1% of total inertia) is correlated to all temperature daily variables (calculated as differences between downstream versus upstream), in particular to the maximum daily temperature difference (Tmax_diff). The second axis (25.3%) discriminates the daily amplitude difference (Range_diff) with the minimum temperature difference (Tmin_diff) (Fig. 7). Results of the RDA show that the water residence time and the impoundment surface explain 95.2% of the PCA structure (time series plotted on the first and second axis).The projection of the site series on these axes shows a strong spreading along the first axis. The dams that had two different measurement years stay within the same range on this first axis (i.e., Fretaz and Champagne) (Fig. 8).

Multiple regression analyses between the temperature variables (median values of

Tmin_diff and Tmax_diff) and the physical characteristics obtained by the RDA (residence time and impoundment surface) resulted in high explanatory power ($R^2 \approx 0.7$). These regressions identified the significant contribution of residence time for Tmin_diff and Tmax_diff, whereas only surface area had a significant contribution for Tmax_diff (Table 4).

Figure 7 and 8 changed

A new table is added

Table 4. Results of multiple linear regressions performed on the 2 indicators Tmin_diff, Tmax_diff using the physical characteristics: i) surface, ii) residence time. Significant pvalue are in bold.

Dependent variable Independent variable physical characteristics R2 standardized coefficient pvalue

Tmax_diff surface 0.72 0,39 0.041 residence time 0.80 0.001

Tmin_diff surface 0.68 -0.13 0.48 residence time 0.80 0.001

"7. Section 3.7 – this section does not provide a synthetic view of any of the data, and the intent of this section is unclear. Suggest removing it entirely."

We agree, and have added new text section 3.7 L 234 to 239 to present a more synthetic view of the data. We hope that we have made the intent more clear.

L 234 Focus on temperature pattern in short period of time in intra-daily temperature variation. Previous text: L 235 239 Looking more specifically on a short period of time (three consecutive days), differences in the diurnal variation of the temperature of the river upstream and downstream of the dam shows that for the first group A, the maximum water temperatures upstream and downstream are close, while the minimum temperature downstream does not return to that of upstream (Fig. 9A). In the second group B the water temperature difference between upstream and downstream are

more important and remain persistent during all the day period (Fig. 9B). Replaced by To further illustrate the different thermal regime effects from our typology analysis, we compare intra-daily temperature variations for a three-day time series in group A (small thermal effect) with group B (large thermal effect; Fig. 9): - In the example of group A (Fig. 9A), the downstream temperature is generally warmer than the upstream temperature (observed difference of 1°C warmer) except for a few hours during the three day sample observation period. The biological benchmark of 22°C is exceeded both upstream and downstream during the day of August 20. The rest of the time, temperatures are below this threshold. From a biological point of view, the duration above the thermal threshold is short, preceded and followed by more favorable temperatures (i.e., the remission period). - In the example of group B (Fig. 9B), the downstream temperature is systematically higher than that of the upstream, with a temperature difference varying between +0.8–2.4°C. The 22°C threshold is exceeded downstream for a cumulative 42 h over the three-day period. August 15 and 16 have downstream temperatures that rarely go below 22°C, leaving no time for thermal remission (return to a temperature that is better tolerated physiologically by fish). At the same time, the upstream part of the stream is maintained at daily temperatures not exceeding this threshold. - Additionally; differences in the diurnal temperature variation upstream and downstream of the dam shows that for group A, the maximum water temperatures are close, whereas the minimum temperature downstream does not return to that of upstream (Fig. 9A). In group B the water temperature difference between upstream and downstream are persistent throughout the diurnal cycle (Fig. 9B). For all sites, by studying the average daily duration with a temperature exceeding 22°C continuously, we can see (Fig. 10): - downstream durations are always greater than or equal to that of the upstream durations, regardless of site typology, - the largest upstream/downstream differences occur in the group B2 group, - group A is generally not affected by an upstream/downstream increase, except for two sites which exhibit a two hour increase.

In addition, we added the following new text in 2.4 Data Analysis to further clarify the point of this section about biological importance of thermal effects.

To assess the potential biological importance of dam thermal effects, we also calculated 1) the number of days that water temperatures were greater than 22°C, and 2) the mean of the maximum daily duration (in hours) where water temperature was greater than 22°C. We chose 22°C as an illustrative threshold known to be a thermal stress benchmark value for salmonids (Elliott and Elliot, 2010; Ojanguren et al., 2001).

Previous text : L 162 (iv) the dam thermal effect considering an arbitrary threshold of 22 °C, with a calculation of the number of days above this threshold.

We also added a new synthetic analysis of intra-daily durations above the defined biological threshold. So, we added this text to the data analysis section:

4. calculation of the number of days above the biological 22°C threshold, and 5. calculation of the average maximum daily duration (in hours) above the biological 22°C threshold.

And we further added a sentence to clarify why the threshold was chosen L346: The threshold temperature of 22 °C known to be a thermal stress benchmark value for salmonids especially for brown trout (Salmo trutta) is also known to be important for the life cycle of aquatic invertebrates (Ward, 1976; Brittain and Salveit, 1989).

"8. Section 3.8 – the arbitrary nature of this analysis provides little insight or direct ecological interpretation. In the discussion the authors correctly indicate that the choice of a 22 degree is actually not arbitrary, but has a basis in that temperatures above this point are generally deleterious to salmonids. Although I think this section could be a valuable contribution by the research, the fragmented presentation leads me to suggest removing it entirely."

Fixed; see above.

"9. In the discussion, the authors talk about different years (hot vs. cool, or wet vs. dry), but none of the analysis really looks into this. I think it is an important point, so would like the authors to explore and quantify this in a reasonable way. "

Response: Fixed with new fig. 2 and fig. 5

"10. In the introduction and discussion, the authors talk about the importance of dam and reservoir size, but don't do any formal analysis. At a basic level, it would seem that correlation or regression of reservoir area, and another analysis with residence time, on the response variables of mean temperature difference, mean difference in maximum temperature, and mean difference in minimum temperature would be an important starting point."

Response: The new statistical analyses (Redundancy analysis, multiple regressions) developed above answer this question.

"11. The discussion of biological effects was quite thorough."

Technical Comments: "1. Many grammatical errors – far more than is appropriate for a scientific reviewer to make edits on, but these need to be addressed before publication."

Fixed.

"2. The citation for Dunham et al. is incomplete, but I applaud investigators for addressing instrument calibration issues, which are often ignored!"

Fixed.

We hope we have satisfactorily replied to your comments and issues, which we believe substantially increased the readability and understanding of this manuscript.

Best regards,

The Authors 

Please also note the supplement to this comment:
https://www.hydrol-earth-syst-sci-discuss.net/hess-2019-136/hess-2019-136-AC1-supplement.pdf

[Figure]

[Figure]

**Fig. 1.** Figure 2. Time-series of water temperature (°C) upstream (blue) and downstream (red) of the dam Fretaz, Veyle stream, respectively in years 2014 and 2016.

[Figure]

**Fig. 2.** Figure 2. Time-series of water temperature (°C) upstream (blue) and downstream (red) of the dam Fretaz, Veyle stream, respectively in years 2014 and 2016.

- A trend line with equation y = 0,0077x + 0,7282 and R² = 0,0031

**Fig. 3.** Figure 5. Relation between daily maximum air temperatures (°C), daily up-stream/downstream temperature differences for all the data available for the study.

[Figure]

**Fig. 4.** Figure 4. Minimum (A) and maximum (B) daily temperatures upstream and downstream of the dams-of-the river (Dompierre site, Veyle stream in 2010; Fretaz site, Veyle stream in 2014). Dashed line is 1:1

**B**

**Fig. 5.** Figure 4. Minimum (A) and maximum (B) daily temperatures upstream and downstream of the dams-of-the river (Dompierre site, Veyle stream in 2010; Fretaz site, Veyle stream in 2014). Dashed line is 1:1

[Figure]

**Fig. 6.** Figure 6. Box-plot distribution (25% - 75 %) of upstream/downstream differences of daily maximum (A) and minimum (B) temperatures for all the time-series studied. (Red lines: 0°C for daily maximum te

**Upstream/downstream differences on daily minimum temperatures**

group (A)  group (B1)  group (B2)

**Fig. 7.** Figure 6. Box-plot distribution (25% - 75 %) of upstream/downstream differences of daily maximum (A) and minimum (B) temperatures for all the time-series studied. (Red lines: 0°C for daily maximum te

**Correlation circle**

Tmin_diff

Tmax_diff

Range_diff

F2 (25.3 %)

F1 74.1 %)

**Fig. 8.** Figure 7. PCA analysis. Correlation circle with temperature as active variables

[Figure]

**Fig. 9.** Figure 8. PCA analysis. Scatterplot of time series. Ellipses are drawn to visualize the groups obtained with the hierarchical cluster analysis

**Mean max hours duration with T > 22°C**

Revel 2016

Peroux 2015

Neuf 2016

Thurignat 2016

Peloux 2016

Dompierre 2010

Champagne 2015

| | A |
|---|---|
| ▲ | B1 |
| ◆ | B2 |
| ------- | line 1:1 |

Montfalconnet 2015

Thuets 2016

Champagne 2009

Fretaz 2016

Fretaz 2014

Caillou 2009

*x-axis:* upstream

*y-axis:* downstream

**Fig. 10.** Figure 10. Mean of the daily maximum duration with T above 22 °C , upstream and downstream each site monitored in the study. A (circles), B1 (triangles), B2 (rhombus) are the groups of sites resulting

---

## Author Comment (AC2) · 15 Jul 2019

Dear Editor and Referees,

Thank you for the quality of your proofreading and comments; they have greatly improved the manuscript. We also appreciate your interest in the subject matter, which we think is of critical importance to managers across France and the world who are dealing with issues of small dam removal and ecological integrity. We believe we have substantially addressed all of the outstanding comments and issues, and we look forward to your second review of the work. All of the referees remarked on the issue of data representativeness, so we will briefly discuss this issue here. Data scarcity (i.e., lack of data across years within sites) is a primary challenge for understanding thermal

effects of small dams, and it is one of the primary reasons that we used a compiled dataset with data from field operators, which we bolstered with our own sampling. We acknowledge that using these two data sources may make reading and understanding a little more difficult, but we believe it enriches the analysis by increasing the number of time series and across-year examples, (though we agree this dataset is probably still insufficient to draw broad conclusions). Hence, we are aware of the issues with the dataset, and we have added text throughout to underscore this issue. However, we feel that the analysis and general results are valid and useful, regardless of data scarcity issues, which every study must deal with. Throughout the manuscript, we have made major revisions based on the referees comments and suggestions. The major changes are: - use of new statistical analysis methods to strengthen the robustness of the results, - improved consistency between points raised in the comments and proposed figures, - grammatical quality review: a final revision of English was done by a native speaker.

General comments: "In general, the paper discusses a relevant research issue, as is discussed based on the literature in the discussion. It is apparently based on an interesting dataset (though with some limitations, mentioned below), but the presentation and discussion of the results is relatively poor and not very clear, and calls for major revisions." "the presentation and discussion of the results is relatively poor"

Response: We have significantly improved the version submitted, adding all the statistical analyses required to support the results. They reinforce, but do not change their meaning.

General comments:"It should be made more clear (in the introduction etc.), that the results are probably not easily transferrable to other areas, as the choses study sites are quite homogenous (focus on a certain region of France). "

Response: While we acknowledge the reviewer's comment that our study is based on a regional dataset, we believe that the results (i.e., that dam physical attributes influence

downstream thermal regimes) is applicable to many other regions and systems. Additionally, we wanted to focus our results on the importance of these thermal regimes on ecophysiological processes, like effects to the brown trout. We have added new text throughout the paper to clarify this point. To remove any ambiguity, we also delete the reference to regional stream temperature model in the abstract (L 12) and the introduction (L 114) On the other hand, we propose in the discussion to complete the notion of the possibility of regionalization as follows

Previous text: L 323 One potential path for deepening research is regionalization as a function of thermal regimes and their governing factors (characteristics of aquifers/climate/bed material/conductivity).

Replaced by

One potential path forward is to create regionalized statistical models based on geographical data and dam databases, analogous to the way that ecological risk analyses are constructed (Allan et al. 2012; Van Looy et al., 2015). However, we realize that our dataset is provincial in temporal and regional extent, potentially limiting extrapolation of results to other areas with different groundwater and climatic influences.

General comments: "Furthermore, the study would greatly benefit from including more temperature data from the same site for several years – one would expect to also see quite some inter-annual differences. As this does not seem to be possible, the authors should at least discuss this shortcoming. Especially as the authors try to hint at a regionalization (e.g. at the end of section 4.1), this should be discussed better: What, for example, about the different groundwater regimes – are we talking about gaining or losing rivers? Etc."

Response: We have added a sentence to the discussion acknowledging these issues. Line 325 However, we realize that our dataset is provincial in temporal and regional extent, potentially limiting extrapolation of results to other areas with different groundwater and climatic influences.

General comments: "The overall result – that the most important drivers of temperature regime changes in dams are residence time and surface area are not particularly surprising. Discuss this. (maybe one could even come up with some empirical linear relationship or empirical model, including those parameters, and water temperature, air temperature, solar radiation etc.?)"

Response: We agree that the results are not particularly surprising, but we note that these results are surprisingly absent from the literature. Hence, this work provides an important result that, to our knowledge, has not been previously presented. We have tried to quantify the heating due to the structures of small dams. The major determining parameters that emerge do not contradict physical knowledge. But it is important to point out that we were not seeking to highlight the physical determinants of the thermal regimes of rivers, but rather the factors responsible for heating due to a dam and its associated impoundment. We have thus provided knowledge on the orders of magnitude of heating for structures that have not yet been well documented. We have added statistical analysis (see later) to explain more efficiently these relationships, and have added text throughout to better address the issues raised in this comment. Sentence added L307 We confirm with the redundancy analysis that residence time and surface area of the water body are the principal explanatory variables of the upstream/downstream temperature differences. However, these relationships are not entirely clear, as the multiple regressions (Table 4) indicate that diff_Tmax is best explained by both residence time and surface area, whereas diff_Tmin is best explained only with residence time. "Specific comments:" "Section 1: Please include some more general explanation on why the whole issue of dams changing the thermal regime is relevant (make your motivation more clear)"

Response: We have clarified the motivation for this study explained in the introduction with an English speaker and hope it addresses this comment (paragraph 1.5, line 87 to 107). We review the literature and show that knowledge is scattered regarding the orders of magnitude of thermal effects that are significant for biological processes. Our

HESSD

goal is therefore to better document these orders of magnitude.

"Line 27: "These determinants are candidate to generalize results" – sentence a bit unclear, please reformulate"

Response: Sentence deleted.

"Line 47: "During summer, the factors leading to warming are: (i) the input of heat from upstream" – maybe you should be a bit more specific here. Mention why you focus on summers. What do you mean by the input of heat from upstream? Tributaries that are warmer than the main stream?" Responses: Focus on summer: We have mainly targeted the biological risk related to global warming. Introduction §1.1 line 35 – 37 "As ectotherms, aquatic organisms are very sensitive to ambient water temperature and to its alteration, especially in the vicinity of their upper thermal temperature tolerance (Brett, 1979; Coutant, 1987; McCullough et al., 2009 for Coldwater fish review; Souchon and Tissot, 2012 for European non salmonid fish review).".

Heat from upstream We refer to the conceptual heat flow balance model of Kelleher et al., 2012: the heat flow from upstream depends on the inflow flow Qi and the temperature of the watercourse, which results from the addition of flows from the main river and its tributaries upstream of the studied section. Kelleher, C., Wagener, T., Gooseff, M., Mcglynn, B., Mcguire, K. and Marshall, L. (2012). Investigating controls on the thermal sensitivity of Pennsylvania streams. Hydrological Processes. 26(5): 771-785. To be precise we add "fluxes" in L 47

"Line 50: If you talk about different anthropogenic influences on stream temperature, you probably also should mention cooling water from power plants etc."

Response: The objective of the study is to quantify the effects small dams in stream; this does not concern cooling water from power plants affecting large rivers.

"Line 56: > 15 m of what?"

Fixed 15 m high

"Line 61 ff: These two "predictions" you are mentioning from 1983 and 1990 should be verified by now? Can you say something about this?"

The term prediction is inappropriate

Fixed

Previous text: L 61 to 63 In addition, Ward and Stanford (1983) predicted that dams in headwaters might not alter the natural temperature range, with the assumption that canopy and springs or groundwater influx can buffer annual temperature variations.

Replaced by

In addition, Ward and Stanford (1983) have made the general assumption that dams in headwaters might not alter the natural temperature range, with the assumption that canopy and springs or groundwater influx can buffer annual temperature variations. Furthermore, SDC mentioned summer water temperature warming downstream of surface reservoir's release (O'Keeffe et al., 1990).

"Line 84: With a height smaller than 5m?"

Fixed L 84 We studied dams with height smaller than 5 m, called hereafter simply small dams.

"Line 88ff: Be more precise here. There are few articles even considering temperature effects? Those are the 43 sites or articles?"

Fixed on 43 studies , 25 % have been considered having a temperature increase effect

"Line 106: "with closed riparian canopy or aquifers" – what do you want to say here?"

Previous text: L105 to 106 This variability is greater in headwaters due to the weak thermal inertia and great diversity of these waterbodies, and also to heterogeneous effects with closed riparian canopy or aquifers.

Replaced by

This variability is greater in headwaters due to the weak thermal inertia and great diversity of these waterbodies, especially with regard to local shading effects from riparian canopy cover and relative importance of spring or tributary discharges.

"Line 106ff: "This is the reason why it seems preferable in a first study to focus on the single effects of the impoundment immediately downstream the dam." – please reformulate/make your motivation more clear. How exactly is this resulting from the above?"

Fixed Previous text L 106 to 107 This is the reason why it seems preferable in a first study to focus on the single effects of the impoundment immediately downstream the dam

Replaced by

Given this potential complexity with several possible confounding factors, the study focused only on the warming effect of small dams and their impoundment. "Line 130: How is a "day of heat wave" defined?"

For scenario A1B (mean concentration of greenhouse gases), the estimation was more than ten additional days of heat waves by 2050.

Response: The definition is conform to International meteorological vocabulary WMO, 1996. WMO, No. 182. TP. 91. Geneva (Secretariat of the World Meteorological Organization) 1966. Pp. xvi, 276. Sw. fr. 40 "Marked warming of the air, or the invasion of very warm air, over a large area; it usually lasts from a few days to a few weeks"

Fixed

Previous text: L129 to 130 For scenario A1B (mean concentration of greenhouse gases), the estimation was more than ten additional days of heat waves by 2050.

Replaced by

For scenario A1B (mean concentration of greenhouse gases), the estimation was more than ten additional days of heat waves (WMO, 1966) by 2050.

"Section 2.2: Mention right away in the text how many dams you study. And how did you chose those specific sites?"

Fixed

L 132 The 11 dams in the study area are overflow structures and . . .

The sites were chosen taking into account their distribution in the upstream downstream gradient and the size gradient of the reservoirs.

Line 145: Make it clear that the temperature sampling was performed for single summers (or two) per site, between 2009 and 2016

Fixed We add sentence: L 146 For two sites, we have series for 2 different summers (Champagne2009 and 2015, Fretas 2002014 and 2016) because the local water management organization was particularly interested in the thermal regimes of these rivers. (Table 1).

"Section 2.5: Please elaborate further on how you performed your PCA. Illustrative variables are explanatory variables? "In order to identify characterization of the impacts of the different dams" – reformulate, unclear!"

Fixed Previous text: L 166 to 170 2.5 PCA analysis In order to identify the characterization of the impacts of the different dams, a principal component analysis (PCA) was carried out using the software XLStat (ADDINSOFT$^{TM}$) on the water temperature variables: downstream / upstream difference of the maximum, average and minimum daily temperature and daily temperature amplitude. The physical characteristics of the structures (Table 1) were used as illustrative variables to evaluate the correlations with the temperature variables

Replaced by

2.5 Ordination analysis To characterize the impacts of the different dams, a principal component analysis (PCA) was carried out using the software XLStat (ADDINSOFT[TM]) on the three water temperature variables: downstream/upstream difference of the maximum and minimum daily temperature and daily temperature range. We used the median values for variables on each time-series in order to build an input matrix (13 occurrences for three variables). Then a complementary redundancy analysis (RDA) with automatic stepwise variable selection procedure was used to identify the physical dam characteristics (Table 1) that significantly explain the PCA results (ter Braak 1986). After the RDA identified the relevant physical dam characteristics, we conducted multiple linear regression between these characteristics and temperature variables to determine specific effect sizes of these characteristics on thermal regime. Ter Braak, C. J. F.: Canonical correspondence analysis: a new eigenvector technique for multivariate direct gradient analysis, Ecology, 67, 1167-1179, 1986.

Previous text : L 220 232 3.6 PCA results The first axis of the PCA analysis (78.3 %) is correlated to all temperature daily variables (calculated as differences between downstream versus upstream), in particular to the maximum daily temperature difference (Tmax_diff). The second axis discriminates the daily amplitude difference (Range_diff) with the minimum temperature (Tmin_diff) difference (Fig. 7). For the determinants, the water residence time is the most correlated variable to the first axis F1, the size of the reservoir (surface, volume, length) correlates to both the first and second axis. The other physical-geographical characteristics related to the size of the watercourse (watershed, distance to the source), are correlated with the daily maximum temperature and associated with the second axis F2 (20.7 %); dam height has a very weak correlation with the axis F1. The projection of the site series on these axes shows a strong spreading along the first axis. The dams measured two different years stay within the same range on this axis (Fretaz and Champagne) (Fig. 8). Groups B1 and B2 are distinguished by respectively the first and second axis association. This can be linked to the determinants of strong residence time influence for group B2, whereas group B1

is mainly characterized by the size of the impoundment (large impoundments, yet with relatively smaller residence time and thus less exacerbated thermal regime effects).

Replaced by

3.6 Ordination results The first axis of the PCA analysis (74.1% of total inertia) is correlated to all temperature daily variables (calculated as differences between downstream versus upstream), in particular to the maximum daily temperature difference (Tmax_diff). The second axis (25.3%) discriminates the daily amplitude difference (Range_diff) with the minimum temperature difference (Tmin_diff) (Fig. 7). Results of the RDA show that the water residence time and the impoundment surface explain 95.2% of the PCA structure (time series plotted on the first and second axis).The projection of the site series on these axes shows a strong spreading along the first axis. The dams that had two different measurement years stay within the same range on this first axis (i.e., Fretaz and Champagne) (Fig. 8).

Multiple regression analyses between the temperature variables (median values of Tmin_diff and Tmax_diff) and the physical characteristics obtained by the RDA (residence time and impoundment surface) resulted in high explanatory power ($R2 \approx 0.7$). These regressions identified the significant contribution of residence time for Tmin_diff and Tmax_diff, whereas only surface area had a significant contribution for Tmax_diff (Table 4).

A new table is added

Table 4. Results of multiple linear regressions performed on the 2 indicators Tmin_diff, Tmax_diff using the physical characteristics: i) surface, ii) residence time. Significant pvalue are in bold.

Dependent variable; Independent variable physical characteristics; R2; standardized coefficient; pvalue

Tmax_diff ;surface; 0.72; 0,39;0.041;

; residence time;;0.80;0.001

Tmin_diff; surface; 0.68; -0.13; 0.48

;residence time; ;0.80;0.001

"Section 3.2/Fig. 4: I understand that the scatter plot for Dompierre shows "type 2", so like in Figure 3. However, Neuf in Fig. 4 does not show "type 1", like in Figure 2, because there is almost no difference between minimum temperatures up- and downstream. And, why don't you simply show the same data in your timeseries plots (Fig. 2 and 3) and the scatterplot (Fig. 4) to illustrate the two types. Also, better to combine the figures and make the two types more clear by that."

Response: We follow the recommendation and propose a new set of figures

Fig. 2 Fretaz 2014 and 2016 and Fig. 4 Dompierre (type 2) and Fretaz (type 1)

Previous text: L 191 to 194 The two dominant patterns can be illustrated by plotting the minimum and maximum temperature values at the site "Dompierre 2010" with a difference of order of + 1.5°C between the upstream and downstream of the site, comparing to "Neuf 2016", where these values are the same for minimum daily temperatures, or even slightly negative for the maximum temperatures (Fig. 4).

Replaced by

The two dominant patterns of temperature differences are further illustrated by plotting the minimum and maximum temperature values at the site. For example, at Dompierre in 2010, we observed a consistent shift of approximately +1.5°C (both maximum and minimum daily temperature) between the upstream and downstream of the dam (Fig. 4A). In contrast, at Fretaz in 2014, this shift is dampened, and temperature values between upstream and downstream follow a 1:1 relationship (Fig. 4B). New figure 4

"Section 3.3: 0.46% of what?"

L 197 This difference averages 0.46% for the 13 cases.

Response: This precision is deleted, as it is secondary

"Section 3.5: Specify how you calculate your differences (downstream – upstream?). And don't groups B1 and B2 both exhibit net warming? Be more precise."

Response: We propose to modify the section 2.4 Data analysis (l 156 à 159)

Previous text: L156 to 159 To determine if the dams alter the temperature regime, the minimum, average and maximum temperatures and amplitudes were calculated for each full day recorded, and the median values were recorded for the period. The calculations of daily differences of maximum and minimum water temperatures were performed for each pair of upstream/downstream records, and the median of these differences over the recording period was calculated.

Replaced by To characterize the influence of dams on stream thermal regimes we first calculated three variables: daily difference between upstream and downstream temperature 1) maximums, 2) minimums, and 3) ranges for each site and year. (..). With these data, we then conducted the following analyses: 1. Median summer differences in maximum, minimum, and range between upstream and downstream (median is used instead of mean to characterize a season in order to limit the effect of a specific weather event), 2. . ..

"Section 3.7: Confusing to speak of "short period of time" or "three consecutive days" – what you actually do is to look at shifts in intra-daily temperature variation."

Fixed

Previous text: L 234 Focus on temperature pattern in short period of time. Replaced by L 234 Focus on temperature pattern in intra-daily temperature variation. Previous text: L 235 239 Looking more specifically on a short period of time (three consecutive days), differences in the diurnal variation of the temperature of the river upstream and downstream of the dam shows that for the first group A, the maximum water temperatures upstream and downstream are close, while the minimum temperature downstream does not return to that of upstream (Fig. 9A). In the second group B the water temperature difference between upstream and downstream are more important and remain persistent during all the day period (Fig. 9B). Replaced by To further illustrate the different thermal regime effects from our typology analysis, we compare intra-daily temperature variations for a three-day time series in group A (small thermal effect) with group B (large thermal effect; Fig. 9): - In the example of group A (Fig. 9A), the downstream temperature is generally warmer than the upstream temperature (observed difference of 1°C warmer) except for a few hours during the three day sample observation period. The biological benchmark of 22°C is exceeded both upstream and downstream during the day of August 20. The rest of the time, temperatures are below this threshold. From a biological point of view, the duration above the thermal threshold is short, preceded and followed by more favorable temperatures (i.e., the remission period). - In the example of group B (Fig. 9B), the downstream temperature is systematically higher than that of the upstream, with a temperature difference varying between +0.8–2.4°C. The 22°C threshold is exceeded downstream for a cumulative 42 h over the three-day period. August 15 and 16 have downstream temperatures that rarely go below 22°C, leaving no time for thermal remission (return to a temperature that is better tolerated physiologically by fish). At the same time, the upstream part of the stream is maintained at daily temperatures not exceeding this threshold. - Additionally; differences in the diurnal temperature variation upstream and downstream of the dam shows that for group A, the maximum water temperatures are close, whereas the minimum temperature downstream does not return to that of upstream (Fig. 9A). In group B the water temperature difference between upstream and downstream are persistent throughout the diurnal cycle (Fig. 9B). For all sites, by studying the average daily duration with a temperature exceeding 22°C continuously, we can see (Fig. 10): - downstream durations are always greater than or equal to that of the upstream durations, regardless of site typology, - the largest upstream/downstream differences occur in the group B2 group, - group A is generally not affected by an upstream/downstream increase, except for two sites which exhibit a two hour increase.

A new sentence is added in 2.4 data analysis To assess the potential biological importance of dam thermal effects, we also calculated 1) the number of days that water temperatures were greater than 22°C, and 2) the mean of the maximum daily duration (in hours) where water temperature was greater than 22°C. We chose 22°C as an illustrative threshold known to be a thermal stress benchmark value for salmonids (Elliott and Elliot, 2010; Ojanguren et al., 2001).

L 162 (iv) the dam thermal effect considering an arbitrary threshold of 22 °C, with a calculation of the number of days above this threshold. Replaced by

4. calculation of the number of days above the biological 22°C threshold, and 5. calculation of the average maximum daily duration (in hours) above the biological 22°C threshold.

And in discussion L 344 to 349 We have chosen temperature > 22°C as an illustrative threshold known to be a thermal stress benchmark value for salmonids especially for brown trout, Salmo trutta (Elliott and Elliot, 2010: upper critical incipient lethal temperature for alevins considered as a very sensitive stage; Ojanguren et al., 2001: general activity of brown trout juvenile). We also know that thermal regime and threshold values are important for the life cycle of aquatic invertebrates (Ward, 1976; Brittain and Salveit, 1989), and it is possible that changes in natural temperature regimes may be as important as altered stream flows to the ecological impacts of dam operations (Olden and Naiman, 2010). Replaced by In this study, we used a temperature of 22°C as an illustrative threshold known to be a thermal stress benchmark value for salmonids, especially for brown trout, Salmo trutta (Elliott and Elliot, 2010: upper critical incipient lethal temperature for juveniles, which is considered a very sensitive stage; Ojanguren et al., 2001: general activity of brown trout juvenile). In addition; this threshold is known to be important for the life cycle of aquatic invertebrates (Ward, 1976; Brittain and Salveit, 1989).

We add a new figure (Fig.10) "Section 4, first paragraph: Some of this would be better

in the introduction. Same applies to first two paragraphs of section 4.1."

Response: That's right. We think that the recall of the context in a few sentences make the discussion as an independently readable part.

"Line 317, 318: Again, specify the sign of your temperature differences."

Fixed L317 in the order of + 0.6 to + 2.4°C

"Line 344ff: Is Salmo trutta a common species in the rivers of your test sites?"

Response: Yes, Salmo trutta is endemic and emblematic and at the ecological limit of his distribution. This is why a warming effect added by dams to the natural thermal regime is likely to further limit its range.

"Line 378: "The thermal landscape is therefore potentially very fragmented due to this fact alone." What do you mean by this and the following sentences?"

Fixed Previous text: L378 The thermal landscape is therefore potentially very fragmented due to this fact alone.

Replaced by

because of the high density of dams in the landscape (0.64 per km), the thermal landscape of this region is potentially fragmented.

"Line 385: Please specify which "spatial generalization elements" you mean."

Fixed Previous text : L384 to 385 Our work provides spatial generalization elements to better document the present and future thermal landscape

Replaced by

Our work highlights physical dam characteristics that could be useful in a large-scale heat risk analysis, or in modeling scenarios aiming to account for changes in thermal regimes.

Technical comments: "Be consistent with thousand separators (for example, you have 2 710000, 96 222, 59071)"

Fixed

"Be consistent on how to write "run-of-the-river dam"."

Fixed

"Line 38: Why do you cite Rader et al., 2007 as part of the review by Ellis and Jones?"

Fixed L38 (Rader et al., 2007 in Ellis and Jones, 2013)

Replaced by

(Rader et al., 2007)

"Line 42: "precipitation", not "precipitations", this comes up several times"

Fixed Lines 42,153, 154

"Line 68: reformulate to "they are expected to increase downstream water temperature" or similar" Fixed Previous text: L68 they are expected to deliver downstream warmer water

Replaced by they are expected to increase downstream water temperature

"Line 78: "(ROE, sept 2017)" why is this cited this way?"

Fixed Suppressed

"Line 59: "water temperature patterns for tens of km"?"

Fixed Previous text: L59 alter longitudinal downstream water temperature pattern tens of km

Replaced by

alter longitudinal downstream water temperature pattern for tens of km
"Line 72ff: "very imprecise depending on national databases. For example, the International Commission on Large Dams""

Fixed Previous text: L 72 nation databases.

Replaced by

national databases.

"Line 90ff: "Dripps et al. (2013): : :." – please reformulate, sentence unclear"

Fixed

Previous text : L90 to 92 Dripps et al. (2013) studying 3 residential artificial headwater lakes (17 to 45 ha) on stream (low flow discharge 0.0024 to 0.0109 m3/s) showed that they could increase summer downstream temperature by as much 8.4°C and decrease diurnal variability by as much 3.9°C.

Replaced by

.Dripps et al. (2013) studied the influence of three residential artificial headwater lakes (17–45 ha) on stream (low flow discharge 0.0024 to 0.0109 m3/s) thermal regimes. They measured a summer downstream temperature increase by as much 8.4°C and a decrease of diurnal variability by as much 3.9°C. "Line 95 ff: "Hayes et al. (2008) in the region of the Great Laurentian Lakes" – all this paragraph contains typos and grammar mistakes, please revise"

Fixed

Previous text: L95 to 97 Hayes et al. (2008) in the region of Great Laurentian Lakes measured a weak to null thermal effect of low-head barriers (<0.5 m in height) built to prevent the upstream migration of sea lamprey Petromyzon marinus, but a temperature elevation comprised between 0.0 to 5.6°C below small hydroelectric dams.

Replaced by

In the region of Great Laurentian Lakes, Hayes et al. (2008) studied two types of dams with different uses. They measured a weak to null thermal effect of low-head barriers (height <0.5 m) built to prevent upstream migration of sea lamprey (Petromyzon marinus, L.). On the other hand, they measured a greater effect for small hydroelectric dams (downstream temperature increases up to 5.6°C).

"Line 101: Maybe "explaining variables" is a better term" Fixed Previous text: L 101 to 102 and the difficulty to identify the master variables governing the thermal regime

Replace by

and the difficulty to identify the explaining variables governing the thermal regime

"Sector 2.1: Please revise language. Remove repetitive "on a basis of 230 000 km streams with permanent flow"""

Fixed

Previous text: L 121 to 123 with a dam and weir density of 0.64 features per km greater than the French average of 0.42 features per km (Référentiel national des Obstacles à l'Ecoulement, ROE, September 2017) on a basis of 230 000 km for streams with permanent flow.

Replaced by

Dam and weir density are 0.64 features per km, which is 50% greater than the French average of 0.42 features per km for streams with permanent flow.

We hope we have satisfactorily replied to your comments and issues, which we believe substantially increased the readability and understanding of this manuscript.

Best regards,

The Authors. 

Please also note the supplement to this comment:
https://www.hydrol-earth-syst-sci-discuss.net/hess-2019-136/hess-2019-136-AC2-supplement.pdf

—————————————————————

**Fig. 1.** Figure 7. PCA analysis. Correlation circle with temperature as active variables

[Figure]

**Fig. 2.** Figure 7. PCA analysis. Correlation circle with temperature as active variables

[Figure]

**Fig. 3.** Figure 2. Time-series of water temperature (°C) upstream (blue) and downstream (red) of the dam Fretaz, Veyle stream, respectively in years 2014 and 2016.

[Figure]

**Fig. 4.** Figure 2. Time-series of water temperature (°C) upstream (blue) and downstream (red)
of the dam Fretaz, Veyle stream, respectively in years 2014 and 2016.

**T daily min**

◆ Dompierre

▲ Fretaz

downstream

upstream

A

**Fig. 5.** Figure 4. Minimum (A) and maximum (B) daily temperatures upstream and downstream of the dams-of-the river (Dompierre site, Veyle stream in 2010; Fretaz site, Veyle stream in 2014). Dashed line is 1:1

[Figure]

**Fig. 6.** Figure 4. Minimum (A) and maximum (B) daily temperatures upstream and downstream of the dams-of-the river (Dompierre site, Veyle stream in 2010; Fretaz site, Veyle stream in 2014). Dashed line is 1:1

[Figure]

**Fig. 7.** Figure 10. Mean of the daily maximum duration with T above 22 °C , upstream and downstream each site monitored in the study. A (circles), B1 (triangles), B2 (rhombus) are the groups of sites resulting

---

## Author Comment (AC3) · 15 Jul 2019

Dear Editor and Referees,

Thank you for the quality of your proofreading and comments; they have greatly improved the manuscript. We also appreciate your interest in the subject matter, which we think is of critical importance to managers across France and the world who are dealing with issues of small dam removal and ecological integrity. We believe we have substantially addressed all of the outstanding comments and issues, and we look forward to your second review of the work. All of the referees remarked on the issue of data representativeness, so we will briefly discuss this issue here. Data scarcity (i.e., lack of data across years within sites) is a primary challenge for understanding thermal

effects of small dams, and it is one of the primary reasons that we used a compiled dataset with data from field operators, which we bolstered with our own sampling. We acknowledge that using these two data sources may make reading and understanding a little more difficult, but we believe it enriches the analysis by increasing the number of time series and across-year examples, (though we agree this dataset is probably still insufficient to draw broad conclusions). Hence, we are aware of the issues with the dataset, and we have added text throughout to underscore this issue. However, we feel that the analysis and general results are valid and useful, regardless of data scarcity issues, which every study must deal with. Throughout the manuscript, we have made major revisions based on the referees comments and suggestions. The major changes are: - use of new statistical analysis methods to strengthen the robustness of the results, - improved consistency between points raised in the comments and proposed figures, - grammatical quality review: a final revision of English was done by a native speaker.

"General comments: The purpose of this study was to quantify the downstream impacts of different types of small dams on summer water temperature in lowland streams. The topic of this manuscript is of high importance, and the research is critically needed since water temperature could impact the structure of aquatic communities and the functioning of the aquatic ecosystem as stated by the authors. The data set on water temperature the authors have collected seems to be robust, and with quite enough number of sites. I personally appreciated the calibration process made for the instruments to insure reliable data. The discussion is quite thorough and insightful, but more focus on literature review (others work) rather than focusing on the discussion of the current work. I found that data analysis severely lacking, and the presentation of the results to be using individual sites as examples that are difficult to judge if they are really representative. Therefore, without adequate data analysis I felt that the conclusions were not well supported. The language used is not sufficiently comprehensible and needs to be improved before publication. Many other specific and technical comments can be found below."

Response: We have taken all these comments into account and paid particular attention to the statistical analysis of the data to support our conclusions.

Specific comments "1. P5, L159: Why authors calculate median differences and not mean? Please justifying why this metric instead of means."

Response: We prefer to work with seasonal variables that are not affected by exceptional one-time weather events.

To avoid any confusion, we eliminate any reference to daily mean temperature And we propose to modify the section 2.4 Data analysis (l 156 à 159)

Previous text: L156 to 159 To determine if the dams alter the temperature regime, the minimum, average and maximum temperatures and amplitudes were calculated for each full day recorded, and the median values were recorded for the period. The calculations of daily differences of maximum and minimum water temperatures were performed for each pair of upstream/downstream records, and the median of these differences over the recording period was calculated.

Replaced by

To characterize the influence of dams on stream thermal regimes we first calculated three variables: daily difference between upstream and downstream temperature 1) maximums, 2) minimums, and 3) ranges for each site and year. (..). With these data, we then conducted the following analyses: 1. Median summer differences in maximum, minimum, and range between upstream and downstream (median is used instead of mean to characterize a season in order to limit the effect of a specific weather event), 2. ...

"2. Section 3.5: What is the scientific method used for group clustering?"

Fixed We add description of the statistical method used

Previous text: L 164 Finally, we propose a classification of the observed thermal behavior in 3 groups, based on differences between upstream and downstream dam daily maximum temperature, daily minimum temperature and daily amplitudes.

Replaced by 2.5 Site typology analysis We observed different thermal regimes in our data and wanted to classify them. To do so, we carried out a hierarchical cluster analysis using Euclidian dissimilarities matrix according to the Ward's method (1963) using daily dataset (n=807) of upstream/downstream differences between maximum and minimum temperatures obtained over all time-series. We forced the classification to integrate the different time-series effect by adding a complete disjunctive table differentiating each time-series to the data set. This procedure makes it possible to group the data first by time-series, then in a second step to differentiate them from each other (i.e., to differentiate site thermal regimes).

Previous text: L 204 to 217 3.5 Site typology based on summer thermal regime The median values of the daily temperature variables calculated over summer (from 01/07 to 01/09) permit distinguishing two major types of response to the presence of a small dam (Table 3). A first group (A) is characterised by: - a median of the differences upstream/downstream of the maximum daily temperatures lower than 0.5°C; - a median of the differences upstream/downstream of the minimum daily temperatures between + 0.4 and 1.3°C; - a median of the differences in daily amplitudes lower than - 0.2°C. A second group (B) is characterised by: - a median of the differences upstream/downstream of the maximum daily temperatures higher than 0.5°C; - medians of the differences upstream/downstream of the maximum and minimum daily temperatures in the same order of amplitude. In addition two subgroups can be distinguished: subgroup (B2) with medians of upstream/downstream differences of daily maximum and minimum temperatures higher than 1°C, i.e. net warming between upstream and downstream, and subgroup (B1) with values ranging from 0.3 – 0.8°C.

Replaced by 3.5 Site typology The hierarchical cluster analysis applied on the values of the daily temperature variable differences over summer (from 1 July to 31 August) distinguished three groups: - a first group (A) characterized by: - a median of the differences upstream/downstream of the maximum daily temperatures less than 0.5°C;
- a median of the differences upstream/downstream of the minimum daily temperatures between + 0.4–1.3°C; - a median of the differences in daily amplitudes less than -0.2°C. - a second group (B1) characterized by: - a median of the differences upstream/downstream of the maximum daily temperatures ranging from +0.6–1.2 °C; - a median of the differences upstream/downstream of the minimum daily temperatures between +0.3–1.1°C. - a third group (B2) is characterized by medians of upstream/downstream differences of daily maximum and minimum temperatures both higher than 1.2 °C (i.e., net warming between upstream and downstream)

Figure 6 changed

"3. Section 3.7: the results presented in this section are unclear and the purpose of presenting such results is unclear as well. I found it very hard to link this section with the discussion section. This would be easy for the reader if the results and discussion section were compiled in one section."

Fixed Fully rewritten

Previous text L234 to 239 3.7 Focus on temperature pattern in short period of time Looking more specifically on a short period of time (three consecutive days), differences in the diurnal variation of the 235 temperature of the river upstream and downstream of the dam shows that for the first group A, the maximum water temperatures upstream and downstream are close, while the minimum temperature downstream does not return to that of upstream (Fig. 9A). In the second group B the water temperature difference between upstream and downstream are more important and remain persistent during all the day period (Fig. 9B).

Replaced by 3.7 Focus on temperature pattern in intra-daily temperature variations To further illustrate the different thermal regime effects from our typology analysis, we compare intra-daily temperature variations for a three-day time series in group A (small thermal effect) with group B (large thermal effect; Fig. 9): - In the example of group

A (Fig. 9A), the downstream temperature is generally warmer than the upstream temperature (observed difference of 1°C warmer) except for a few hours during the three day sample observation period. The biological benchmark of 22°C is exceeded both upstream and downstream during the day of August 20. The rest of the time, temperatures are below this threshold. From a biological point of view, the duration above the thermal threshold is short, preceded and followed by more favorable temperatures (i.e., the remission period). - In the example of group B (Fig. 9B), the downstream temperature is systematically higher than that of the upstream, with a temperature difference varying between +0.8–2.4°C. The 22°C threshold is exceeded downstream for a cumulative 42 h over the three-day period. August 15 and 16 have downstream temperatures that rarely go below 22°C, leaving no time for thermal remission (return to a temperature that is better tolerated physiologically by fish). At the same time, the upstream part of the stream is maintained at daily temperatures not exceeding this threshold. - Additionally; differences in the diurnal temperature variation upstream and downstream of the dam shows that for group A, the maximum water temperatures are close, whereas the minimum temperature downstream does not return to that of upstream (Fig. 9A). In group B the water temperature difference between upstream and downstream are persistent throughout the diurnal cycle (Fig. 9B). For all sites, by studying the average daily duration with a temperature exceeding 22°C continuously, we can see (Fig. 10): - downstream durations are always greater than or equal to that of the upstream durations, regardless of site typology, - the largest upstream/downstream differences occur in the group B2 group, - group A is generally not affected by an upstream/downstream increase, except for two sites which exhibit a two hour increase.

A new sentence is added in 2.4 data analysis To assess the potential biological importance of dam thermal effects, we also calculated 1) the number of days that water temperatures were greater than 22°C, and 2) the mean of the maximum daily duration (in hours) where water temperature was greater than 22°C. We chose 22°C as an illustrative threshold known to be a thermal stress benchmark value for salmonids (Elliott and Elliot, 2010; Ojanguren et al., 2001).

L 162 (iv) the dam thermal effect considering an arbitrary threshold of 22 °C, with a calculation of the number of days above this threshold.

Replaced by 4. calculation of the number of days above the biological 22°C threshold, and 5. calculation of the average maximum daily duration (in hours) above the biological 22°C threshold.

L 344 to 346 placed in data analysis L 164 We have chosen temperature > 22°C as an illustrative threshold known to be a thermal stress benchmark value for salmonids especially for brown trout, Salmo trutta (Elliott and Elliot, 2010: upper critical incipient lethal temperature for alevins considered as a very sensitive stage; Ojanguren et al., 2001: general activity of brown trout juvenile).

Previous text: L 344 to 346 We have chosen temperature > 22°C as an illustrative threshold known to be a thermal stress benchmark value for salmonids especially for brown trout, Salmo trutta (Elliott and Elliot, 2010: upper critical incipient lethal temperature for alevins considered as a very sensitive stage; Ojanguren et al., 2001: general activity of brown trout juvenile).

Replaced by In this study, we used a temperature of 22°C as an illustrative threshold known to be a thermal stress benchmark value for salmonids, especially for brown trout, Salmo trutta (Elliott and Elliot, 2010: upper critical incipient lethal temperature for juveniles, which is considered a very sensitive stage; Ojanguren et al., 2001: general activity of brown trout juvenile). In addition; this threshold is known to be important for the life cycle of aquatic invertebrates (Ward, 1976; Brittain and Salveit, 1989).

L 242 For example, for the maximum daily temperature threshold of 22°C (arbitrary value),

Replaced by

For example, for the maximum daily temperature threshold of 22°C,

"4. P7, section 3.8: Authors mention that the maximum daily temperature threshold

of 22 °C is arbitrary value. While later in the discussion, the authors indicate that the choice of a 22°C is actually not arbitrary. I suggest that authors delete the word arbitrary and explain the basis of this threshold choice."

Fixed Arbitrary is suppressed See above

"5. P8, L255: the authors mention warmer, drier, colder and wetter years. Please discuss how these classifications are made?"

Fixed Clarification by adding a sentence L 153

The summer climatic characteristics for our analysis period are compared with the normal values produced by Meteo France (1981–2010).

"6. P18: Fig.4: what is the reason for comparing temperature of different sites (Dompierre and Neuf) in different years (e.g. 2010 and 2016)."

Fixed

Response: Figure 4 has been modified. We now use the same sites as in Figures 2 and 3 to make it easier to read. The purpose of the comparison is to illustrate the distribution of the differences in diff_Tmin and diff_Tmax between the two main types of thermal response. We follow the recommendation and propose a new set of figures (Fig.2 and Fig.4)

Previous text: L 191 to 194 The two dominant patterns can be illustrated by plotting the minimum and maximum temperature values at the site "Dompierre 2010" with a difference of order of + 1.5°C between the upstream and downstream of the site, comparing to "Neuf 2016", where these values are the same for minimum daily temperatures, or even slightly negative for the maximum temperatures (Fig. 4).

Replaced by

The two dominant patterns of temperature differences can be further illustrated by plotting the minimum and maximum temperature values at the site. For example, at

"Dompierre 2010", we observed a consistent shift of approximately + 1.5°C (both maximum and minimum daily temperature) between the upstream and downstream of the dam (Fig. 4). In contrast, at "Fretaz 2014", this shift is dampened, and temperature values between upstream and downstream follow a 1:1 relationship (Fig. 4).

"7. P19: Fig.3 caption: the authors state "time-series of water temperatures upstream (blue line) and downstream (red line) of the dams of Dompierre and Peroux, Veyle stream (2010 and 2015, two warm summer years, respectively + 1.1°C and 2°C, Table 2)", but when looking back in table 2, I have seen that air temperature difference from normal in 2010 is very small (+ 0.3) and NOT +1.1. The +1.1°C air temperature difference from normal is in the year 2009. Therefore, 2009 is almost four times warmer than 2010, hence one may expect the comparison between 2009 and 2015 instead of 2010 and 2015?"

Fixed Corrected legend and site changed Removal of "two warm summer years, respectively + 1.1°C and 2°C, Table 2" in Fig.3 caption.

"8. P19: Fig.3: Since air temperature difference from normal in 2010 is very small (+ 0.3), why the difference between upstream and downstream water temperature at Dompierre dam is very high? This cannot be due to long residence time and average surface are in absence of warm condition, so what could be the reason/s?"

Response: The low deviation from normal indicates a summer temperature close to this normal. The figure shows that the amount of heat supplied to the stream during a "normal" summer is sufficient to vary the temperature between the upstream and downstream of the dam taking into account the long residence time (8.4 days) and the surface of the water body (10900 m$^2$).

"9. It is insecurely to compare 2014 (cold and wet year) with 2015 (warm and dry year) for at least one site (e.g. Dompierre dam) to see the effect of air temperature."

The difference between the upstream and downstream of the dam does not appear to

be solely related to air temperature, as shown in Figure 5. Unfortunately, we have no data available for the same site for these two years. We modify Figure 5 and the text as follows:

Previous text L 200 to 204 During the summer season, the differences in the daily mean temperatures upstream / downstream, are close or staggered during all the season. It is notable that the variability of the summer air temperature is much higher (range 17°C) than stream temperature (range 7.5°C) for these examples (Fig. 5), and that the daily water temperature is not well correlated to air temperature.

Replaced by

During the summer season, the upstream/downstream daily maximum water temperature differences are not well correlated with air temperature for the same periods. For example, a simple linear regression between daily maximum air temperature and daily maximum water temperature differences indicates that air temperature explains only 0.3% of the variability in upstream/downstream thermal regime shifts (Fig. 5).

Technical corrections: "1. P18: in Fig.2 caption, what is the word "respectively" refer to?"

Fixed Response: New figure with site Fretaz 2014 – 2016 (Fig.2) "respectively" is suppressed

2. P1, L18-19: "The mean increase of the minimum daily temperature was 1°C, with 85 % of the time-series showing an increase > 0.5 °C", this sentence is not clear or grammatically incorrect. Fixed Previous text: L18 to 19 The mean increase of the minimum daily temperature was 1°C, with 85 % of the time-series showing an increase > 0.5 °C.

Replaced by

Across all time series, the mean increase of the minimum daily temperature was 1°C, and for 85% of the sites the increase was higher than 0.5°C. "3. P2, L63-64: "surface

release reservoirs", should read "surface reservoirs' release"."

Fixed

"4. P5, L148-149: "in the main flow of the channel" should read "in the main flow channel"."

Fixed

"5. P5, L151: "method Dunham et al. (2005)." should read "method introduced by Dunham et al. (2005)"."

Fixed

"6. P5, L157: the authors state that "and the median values were recorded for the period", how do you record the median? It should read "calculated" instead." Fixed

"7. P6, L182: "Furthermore, the average temperature downstream of the structure was systematically higher or equivalent than that measured upstream" should read "Furthermore, the average temperature downstream of the structure was systematically equivalent or higher than that measured upstream"." Fixed

"8. These are limited examples and the paper contains more. All grammatical errors should be fixed before publication."

A final revision of English was done by a native speaker

We hope we have satisfactorily replied to your comments and issues, which we believe substantially increased the readability and understanding of this manuscript.

Best regards,

The Authors

Please also note the supplement to this comment:
https://www.hydrol-earth-syst-sci-discuss.net/hess-2019-136/hess-2019-136-AC3-

supplement.pdf

[Figure]

**Fig. 1.** Figure 6. Box-plot distribution (25% - 75 %) of upstream/downstream differences of daily maximum (A) and minimum (B) temperatures for all the time-series studied. (Red lines: 0°C for daily maximum te

[Figure]

**Fig. 2.** Figure 6. Box-plot distribution (25% - 75 %) of upstream/downstream differences of daily maximum (A) and minimum (B) temperatures for all the time-series studied. (Red lines: 0°C for daily maximum te

[Figure]

**Mean max hours duration with T > 22°C**

Fig. 3. Figure 10. Mean of the daily maximum duration with T above 22 °C , upstream and downstream each site monitored in the study. A (circles), B1 (triangles), B2 (rhombus) are the groups of sites resulting

[Figure]

**Fig. 4.** Figure 2. Time-series of water temperature (°C) upstream (blue) and downstream (red) of the dam Fretaz, Veyle stream, respectively in years 2014 and 2016.

[Figure]

**Fig. 5.** Figure 2. Time-series of water temperature (°C) upstream (blue) and downstream (red) of the dam Fretaz, Veyle stream, respectively in years 2014 and 2016.

[Figure]

The scatter plot shows:

**T daily min**

- ◆ Dompierre
- ▲ Fretaz

X-axis: upstream (14 to 28)
Y-axis: downstream (14 to 28)

**A**

**Fig. 6.** Figure 4. Minimum (A) and maximum (B) daily temperatures upstream and downstream of the dams-of-the river (Dompierre site, Veyle stream in 2010; Fretaz site, Veyle stream in 2014). Dashed line is 1:1

[Figure]

**Fig. 7.** Figure 4. Minimum (A) and maximum (B) daily temperatures upstream and downstream of the dams-of-the river (Dompierre site, Veyle stream in 2010; Fretaz site, Veyle stream in 2014). Dashed line is 1:1

$$y = 0{,}0077x + 0{,}7282$$
$$R^2 = 0{,}0031$$

**Fig. 8.** Figure 5. Relation between daily maximum air temperatures (°C), daily upstream/downstream temperature differences for all the data available for the study.

---

## Author Response (AR1)

hess-2019-136
Submitted on 28 Mar 2019

Determinants of thermal regime influence of small dams
André Chandesris, Kris Van Looy, and Yves Souchon

Manuscript Type: Research article

Status: Major Revision

**Cover letter**

Dear Editor and Referees,

Thank you for the quality of your proofreading and comments; they have greatly improved the manuscript. We also appreciate your interest in the subject matter, which we think is of critical importance to managers across France and the world who are dealing with issues of small dam removal and ecological integrity. We believe we have substantially addressed all of the outstanding comments and issues, and we look forward to your second review of the work.

All of the referees remarked on the issue of data representativeness, so we will briefly discuss this issue here. Data scarcity (i.e., lack of data across years within sites) is a primary challenge for understanding thermal effects of small dams, and it is one of the primary reasons that we used a compiled dataset with data from field operators, which we bolstered with our own sampling. We acknowledge that using these two data sources may make reading and understanding a little more difficult, but we believe it enriches the analysis by increasing the number of time series and across-year examples, (though we agree this dataset is probably still insufficient to draw broad conclusions). Hence, we are aware of the issues with the dataset, and we have added text throughout to underscore this issue. However, we feel that the analysis and general results are valid and useful, regardless of data scarcity issues, which every study must deal with.

Throughout the manuscript, we have made major revisions based on the referees comments and suggestions. The major changes are:

-       use of new statistical analysis methods to strengthen the robustness of the results,
-       improved consistency between points raised in the comments and proposed figures,
-       grammatical quality review: a final revision of English was done by a native speaker.

**point-by-point response to the reviews**

Below we respond to *comments*, which *are in italics*, in blue, show old text in red, and replacement text in green.

Referee #1
*General comments :*
*" the presentation of the results to be mainly using individual sites as examples that are difficult to judge if they are representative."*

Response: An improvement in the presentation and choice of sites selected as examples has been modified in the final text.

*Specific comments :*
*"1. Figure 2 – why present years in reverse chronological order? Also, why this stream and these years? If possible, it would be preferable to compare 2014 (cold wet year) with 2015 (warmest, dry year in data set)."*

Response: The aim was to highlight that the same site presented the same "patterns" of summer time-series for different years, regardless of the climatic characteristics of the year.
Taking into account the observation, we propose another example (new Figure 2) comparing a cold and humid year (2014) with a normal and dry year (2016) at another site (Veyle stream, Fretaz site): the structure of the thermal patterns between upstream and downstream is preserved.

[Figure]

Figure 2. Time-series of water temperature (°C) upstream (blue) and downstream (red) of the dam Fretaz, Veyle stream, respectively in years 2014 and 2016.

The new text L180 to 189 is modified as

Previous text: L180 to 189
These periods vary from one year to another, likewise the intensity of the increases, but the general pattern remains the same, as demonstrated by the case of the dam Champagne (Renon stream), monitored in 2009 and 2015 (Fig. 2).
Furthermore, the average temperature downstream of the structure was systematically higher or equivalent than that measured upstream.
Different types of time-series were observed regarding the difference between upstream and downstream temperatures:
The most frequent (7/13) is the type observed on the dam of Champagne (Renon stream) in 2009 and 2015; the minimum 185 daily temperatures (T min) are, most usually, higher downstream of the structure, but the maximum daily temperatures (T max) remain within the same magnitudes (Fig. 2, only one example is presented here).
In the other cases (6/13), both the minimum and maximum daily temperatures are higher downstream of the structure, which results in a homothetic lag between the two temperature time-series (Fig. 3).

Replaced by

New text: L199 to 207
The periods of progressively increasing temperature vary in length, magnitude, and timing from one year to another, but the general pattern remains the same, as demonstrated by the case of the Fretaz dam , monitored in 2014 (a cold and humid year) and 2016 (a more normal year, Fig. 2; Table 2).
We observed two consistent patterns in upstream-downstream thermal regimes. In the first pattern, $T_{min}$ is higher downstream, but $T_{max}$ stays relatively constant (Fig. 2). We note that these upstream-downstream differences were muted in 2014, the cold and humid year (Fig. 2). This thermal pattern (i.e., where $T_{min}$ increases downstream, but not $T_{max}$) is observed in 7 out of 13 cases (Table 3). In the other cases (6 out of 13; Table 3), we observed a second pattern, where both $T_{min}$ and $T_{max}$ are higher downstream of the structure, which results in a consistent shift between the two temperature time-series (Fig. 3, selected examples: Dompierre dam 2010 and Peroux dam 2015).

*"2. General – figures don't do a very good job of illustrating points made in text in results. I question whether all the figures are needed (e.g., Figure 3).*

Response: Fixed see above

*Figure 5 – presenting time-series does not show correlation between two variables –one would need to plot air temp vs. water temp to show directly.*

Response: We modify Figure 5 and the text as follows:

Previous text: L200 to 204
During the summer season, the differences in the daily mean temperatures upstream / downstream, are close or staggered during all the season. It is notable that the variability of the summer air temperature is much higher (range 17°C) than stream temperature (range 7.5°C) for these examples (Fig. 5), and that the daily water temperature is not well correlated to air temperature.

Replaced by

New text: L217 to 219
During the summer season, the upstream/downstream daily maximum water temperature differences are not well correlated with air temperature for the same periods. For example, a simple linear regression between daily maximum air temperature and daily maximum water temperature differences indicates that air temperature explains only 0.8% of the variability in upstream/downstream thermal regime shifts (Fig. 5).

[Figure]

Figure 5. Relation between daily maximum air temperatures (°C), daily upstream/downstream temperature differences for all the data available for the study.

*"Figure 4 – never covered in results section."*

Response: We previously covered figure 4 in section 3.2 but now we changed the text to better explain the observed pattern. We also changed the site "Neuf" to "Fretaz 2014".

Previous text: L 191 to 194
The two dominant patterns can be illustrated by plotting the minimum and maximum temperature values at the site "Dompierre 2010" with a difference of order of + 1.5°C between the upstream and downstream of the site, comparing to "Neuf 2016", where these values are the same for minimum daily temperatures, or even slightly negative for the maximum temperatures (Fig. 4).

Replaced by

New text: L209 to 212
The two dominant patterns of temperature differences are further illustrated by plotting downstream versus upstream $T_{min}$ and $T_{max}$ values at the site. For example, at Dompierre in 2010, we observed a consistent shift of approximately +1.5°C (both $T_{min}$ and $T_{max}$) between the upstream and downstream of the dam (Fig. 4A). In contrast, at Fretaz in 2014, this shift is dampened, and temperature values between upstream and downstream more closely follow a 1:1 relationship (Fig. 4B).

New figure 4

[Figure]

Figure 4. Minimum (A) and maximum (B) daily temperatures upstream and downstream of the dams-of-the river (Dompierre site, Veyle stream in 2010; Fretaz site, Veyle stream in 2014). Dashed line is 1:1 line.

*"3. The authors mention differences in mean temperature, but never provide this information in a table. Further, they report median differences without justifying why this metric instead of means. I feel medians can be a useful indicator of central tendency, but the mean is also useful, and needs to be presented if it is discussed."*

Response: To avoid any confusion, we eliminate any reference to daily mean temperature. We also have modified the section 2.4 Data analysis to remove any confusion about using mean temperature (L 156 to 159)

Previous text: L156 to 159
To determine if the dams alter the temperature regime, the minimum, average and maximum temperatures and amplitudes were calculated for each full day recorded, and the median values were recorded for the period. The calculations of daily differences of maximum and minimum water temperatures were performed for each pair of upstream/downstream records, and the median of these differences over the recording period was calculated.

Replaced by

New text: L165 to 170
To characterize the influence of dams on stream thermal regimes we first calculated three variables: daily difference between upstream and downstream temperature 1) maximums ($\Delta T_{max}$), 2) minimums ($\Delta T_{min}$), and 3) amplitudes ($\Delta T_{amp}$) for each site and year. With these data, we then conducted the following analyses:
1.       Median summer differences in $\Delta T_{max}$, $\Delta T_{min}$, and $\Delta T_{amp}$ (median is used instead of mean to limit the influence of extreme values),
2….

*"Section 3.4 – authors state that air and water temperatures do not correlate, but did not perform a correlation analysis".*

Response: Fixed with a new figure 5

*"5. Section 3.5 – how were these groups distinguished (meaning, what formal method was used). My impression is that the investigators did this "by eye", which is not acceptable in my view. A formal cluster analysis would be much more appropriate. Moreover, I think it is hard to defend splitting out groups with such a small number of sites."*

Response: The requested additional statistical analysis has been completed and we propose the following changes

We add description of the statistical method used
Previous text: L 164 to 165
Finally, we propose a classification of the observed thermal behavior in 3 groups, based on differences between upstream and downstream dam daily maximum temperature, daily minimum temperature and daily amplitudes.

Replaced by

New text: L177 to 183
2.5 Site typology analysis
We observed different thermal regimes in our data and wanted to classify them. To do so, we carried out a hierarchical cluster analysis using Euclidian dissimilarities matrix according to the Ward's method (1963) using the daily dataset (n=807) of $\Delta T_{max}$ and $\Delta T_{min}$ obtained over all time-series. We forced the classification to integrate the different time-series effect by adding a complete disjunctive table differentiating each time-series to the data set. This procedure makes it possible to group the data first by time-series, then in a second step to differentiate them from each other (i.e., to differentiate site thermal regimes).

Previous text: L 204 to 217
3.5     Site typology based on summer thermal regime
The median values of the daily temperature variables calculated over summer (from 01/07 to 01/09) permit distinguishing two major types of response to the presence of a small dam (Table 3).
A first group (A) is characterised by:
-       a median of the differences upstream/downstream of the maximum daily temperatures lower than 0.5°C;
-       a median of the differences upstream/downstream of the minimum daily temperatures between + 0.4 and 1.3°C;
-       a median of the differences in daily amplitudes lower than - 0.2°C.
A second group (B) is characterised by:
-       a median of the differences upstream/downstream of the maximum daily temperatures higher than 0.5°C;
-       medians of the differences upstream/downstream of the maximum and minimum daily temperatures in the same order of amplitude.
In addition two subgroups can be distinguished: subgroup (B2) with medians of upstream/downstream differences of daily maximum and minimum temperatures higher than 1°C, i.e. net warming between upstream and downstream, and subgroup (B1) with values ranging from 0.3 – 0.8°C.
Replaced by

New text: L221 to 231
3.5     Site typology

The hierarchical cluster analysis applied to the daily summer temperature anomalies distinguished three groups:
- a first group (A) characterized by :
    - median of $\Delta T_{max}$ less than 0.5°C;
    - median of $\Delta T_{min}$ between + 0.4–1.3°C;
    - median of $\Delta T_{amp}$ less than -0.2°C.
- a second group (B1) characterized by:
    - median of $\Delta T_{max}$ ranging from +0.6–1.2 °C;
    - median of $\Delta T_{min}$ between +0.3–1.1°C.
- a third group (B2) characterized by:
    - median of $\Delta T_{max}$ greater than 1.2 °C;
    - median of $\Delta T_{min}$ greater than 1.2 °C

Figure 6 changed.

[Figure]

Figure 6. Box-plot distribution (25% - 75 %) of upstream/downstream differences of daily maximum and minimum temperatures for all the time-series studied. (Red lines: 0°C for daily maximum temperature and 1°C for daily minimum temperature are drawn to help reading). The vertical lines drawn in bold are the limits to the three classes of results of the hierarchical cluster analysis. Dendrogram CAH's result is shown at the top left of the figure.

*"6. Section 3.6 – in the methods, the authors state that they used mean temperatures in the PCA analysis, but this doesn't show up in the results. Further, the reporting of the PCA results is very incomplete. Loadings of the various variables is needed, as is some criterion for determining what are the significant correlations. I can't say I understand fully how to interpret the circle correlation plot."*

Fixed

Previous text: L 166 to 170    2.5 PCA analysis

[revised manuscript text omitted]

Figure 7 deleted
The new Figure 7 replaces Figure 8

Figure 8. PCA analysis. Correlation circle with temperature as active variables

[Figure]

Figure 7. PCA analysis. Scatterplot of time series. Ellipses are drawn to visualize the groups obtained with the hierarchical cluster analysis

A new table is added
New text: L602 to 604
Table 4. Results of multiple linear regressions performed on the 2 indicators ΔTmin, ΔTmax using the dam physical characteristics surface area and residence time. Significant p-value are in bold.

| Dependent variable | Independent variable physical characteristics | standardized coefficient | p-value | R2 |
|---|---|---|---|---|
| ΔTmax | surface area | 0.39 | 0.041 | 0.72 |
| | residence time | 0.80 | 0.001 | |
| ΔTmin | surface area | -0.13 | 0.48 | 0.68 |
| | residence time | 0.80 | 0.001 | |

*"7. Section 3.7 – this section does not provide a synthetic view of any of the data, and the intent of this section is unclear. Suggest removing it entirely."*

New text section 3.7

Previous text: L 234 to 239
3.7 Focus on temperature pattern in short period of time
Looking more specifically on a short period of time (three consecutive days), differences in the diurnal variation of the temperature of the river upstream and downstream of the dam shows that for the first group A, the maximum water temperatures upstream and downstream are close, while the minimum temperature downstream does not return to that of upstream (Fig. 9A). In the second group B the water temperature difference between upstream and downstream are more important and remain persistent during all the day period (Fig. 9B).

Replaced by
New text: L244 to 265
3.7      Ecologically relevant intra-daily temperature variation
To further illustrate the different thermal regime effects from our typology analysis, we compare intra-daily temperature variations for a three-day time series in group A (small thermal effect) with group B (large thermal effect; Fig. 9):
-        In the example of group A (Fig. 9A), the downstream temperature is generally warmer than the upstream temperature (observed difference of 1°C warmer) except for a few hours during the three day sample observation period. The biological benchmark of 22°C is exceeded both upstream and downstream during the day of August 20. The rest of the time, temperatures are below this threshold. From a biological point of view, the duration above the thermal threshold is short, preceded and followed by more favorable temperatures (i.e., the remission period).
-        In the example of group B (Fig. 9B), the downstream temperature is systematically higher than that of the upstream, with a temperature difference varying between +0.8–2.4°C. The 22°C threshold is exceeded downstream for a cumulative 42 h over the three-day period. August 15 and 16 have downstream temperatures that rarely go below 22°C, leaving no time for thermal remission (return to a temperature that is better tolerated physiologically by fish). At the same time, the upstream part of the stream is maintained at daily temperatures not exceeding this threshold.
-        Additionally; differences in the diurnal temperature variation upstream and downstream of the dam shows that for group A, the maximum water temperatures are similar, whereas the minimum temperature downstream does not return to that of upstream (Fig. 9A). In group B the water temperature difference between upstream and downstream are persistent throughout the diurnal cycle (Fig. 9B).
For all sites, by studying the average daily duration with a temperature exceeding 22°C continuously, we can see (Fig. 10):
-        downstream durations are always greater than or equal to that of the upstream durations, regardless of site typology,
-        the largest upstream/downstream differences occur in the group B2 group,
-        group A is generally not affected by an upstream-downstream increase, except for two sites which exhibit a two hour increase.

Previous text: L 162

(iv) the dam thermal effect considering an arbitrary threshold of 22 °C, with a calculation of the number of days above this threshold.

Replaced by

New text: L173 to 176

To assess the potential biological importance of dam thermal effects, we also calculated 1) the number of days that water temperatures were greater than 22°C, and 2) the mean of the maximum daily duration (in hours) where water temperature was greater than 22°C. We chose 22°C as an illustrative threshold known to be a thermal stress benchmark value for salmonids (Elliott and Elliot, 2010; Ojanguren et al., 2001).

We added a sentence L346

New text: L375 to 376

In addition, this threshold is known to be important for the life cycle of aquatic invertebrates (Ward, 1976; Brittain and Salveit, 1989).

[Figure]

Figure 9. Mean of the daily maximum duration with T above 22 °C , upstream and downstream each site monitored in the study. A (circles), B1 (triangles), B2 (rhombus) are the groups of sites resulting from HCA.

*"8. Section 3.8 – the arbitrary  nature of this analysis provides little insight or direct ecological interpretation. In the discussion the authors correctly indicate that the choice of a 22 degree is actually not arbitrary, but has a basis in that temperatures above this point are generally deleterious to salmonids. Although I think this section could be a valuable contribution by the research, the fragmented presentation leads me to suggest removing it entirely."*

Fixed above

*"9. In the discussion, the authors talk about different years (hot vs. cool, or wet vs. dry), but none of the analysis really looks into this. I think it is an important point, so would like the authors to explore and quantify this in a reasonable way. "*

Response: Fixed with new fig. 2 and fig. 5

*"10. In the introduction and discussion, the authors talk about the importance of dam and reservoir size, but don't do any formal analysis. At a basic level, it would seem that correlation or regression of reservoir area, and another analysis with residence time, on the response variables of mean temperature difference, mean difference in maximum temperature, and mean difference in minimum temperature would be an important starting point."*

Response: The statistical analyses (Redundancy analysis, multiple regressions) developed above answer this question.

*"11. The discussion of biological effects was quite thorough."*

*Technical Comments:*
*"1. Many grammatical errors – far more than is appropriate for a scientific reviewer to make edits on, but these need to be addressed before publication."*

Fixed

*"2. The citation for Dunham et al. is incomplete, but I applaud investigators for addressing instrument calibration issues, which are often ignored!"*

Fixed

Referee #2

*General comments:*

*"In general, the paper discusses a relevant research issue, as is discussed based on the literature in the discussion. It is apparently based on an interesting dataset (though with some limitations, mentioned below), but the presentation and discussion of the results is relatively poor and not very clear, and calls for major revisions."*

*"the presentation and discussion of the results is relatively poor"*

Response: We have significantly improved the version submitted, adding all the statistical analyses required to support the results. They reinforce, but do not change their meaning.

*General comments:"It should be made more clear (in the introduction etc.), that the results are probably not easily transferrable to other areas, as the choses study sites are quite homogenous (focus on a certain region of France). "*

Response: While we acknowledge the reviewer's comment that our study is based on a regional dataset, we believe that the results (i.e., that dam physical attributes influence downstream thermal regimes) is applicable to many other regions and systems. Additionally, we wanted to focus our results on the importance of these thermal regimes on ecophysiological processes, like effects to the brown trout. We have added a sentence in the discussion to clarify this point.

To remove any ambiguity, we delete the reference to regional stream temperature model in the abstract (L 12) and the introduction (L 114)

On the other hand, we propose in the discussion to complete the notion of the possibility of regionalization as follows

Previous text: L 323
One potential path for deepening research is regionalization as a function of thermal regimes and their governing factors (characteristics of aquifers/climate/bed material/conductivity).

Replaced by

New text: L346 to 349
One potential path forward is to create regionalized statistical models based on geographical data and dam databases, analogous to the way that ecological risk analyses are constructed (Allan et al. 2012; Van Looy et al., 2015). However, we realize that our dataset is provincial in temporal and regional extent, potentially limiting extrapolation of results to other areas with different groundwater and climatic influences.

*General comments: "Furthermore, the study would greatly benefit from including more temperature data from the same site for several years – one would expect to also see quite some inter-annual differences. As this does not seem to be possible, the authors should at least discuss this shortcoming. Especially as the authors try to hint at a regionalization (e.g. at the end of section 4.1), this should be discussed better: What, for example, about the different groundwater regimes – are we talking about gaining or losing rivers? Etc."*

Response: We have added a sentence to the discussion acknowledging these issues.
Line 348 to 349

However, we realize that our dataset is provincial in temporal and regional extent, potentially limiting extrapolation of results to other areas with different groundwater and climatic influences.

*General comments: "The overall result – that the most important drivers of temperature regime changes in dams are residence time and surface area are not particularly surprising. Discuss this.* (maybe one could even come up with some empirical linear relationship or empirical model, including those parameters, and water temperature, air temperature, solar radiation etc.?)"*

Response: We agree that the results are not particularly surprising, but we note that these results are surprisingly absent from the literature. Hence, this work provides an important result that, to our knowledge, has not been previously presented. We have tried to quantify the heating due to the structures of small dams. The major determining parameters that emerge do not contradict physical knowledge. But it is important to point out that we were not seeking to highlight the physical determinants of the thermal regimes of rivers, but rather the factors responsible for heating due to a dam and its associated impoundment. We have thus provided knowledge on the orders of magnitude of heating for structures that have not yet been well documented.

But the statistical analysis we performed (see later) explain more efficiently relationships including this parameters.

Sentence added L307 (Previous text)

New text: L328 to 330 and L336 to 338
We found that residence time and surface area were the principal explanatory variables of upstream-downstream temperature differences. Indeed, redundancy analysis indicated the primary differences between our site typologies were explained by these variables.
(…)
Multiple regression (Table 4) clarified the direction and magnitude of these effects and indicated that $\Delta Tmax$ is best explained by both residence time and surface area (group B effects), whereas $\Delta Tmin$ is best explained only with residence time (group A effects).

*"Specific comments:"*
*"Section 1: Please include some more general explanation on why the whole issue of dams changing the thermal regime is relevant (make your motivation more clear)"*

Response: The motivation for the study is explained in the introduction to paragraph 1.5 (line 87 to 107), where we review the literature which shows that knowledge is scattered, and that some of the orders of magnitude characterized are significant for biological processes or organisms by being located in values at risk. Our goal is therefore to better document these orders of magnitude.

*"Line 27: "These determinants are candidate to generalize results" – sentence a bit unclear, please reformulate"*

Response: Sentence deleted.

*"Line 47: "During summer, the factors leading to warming are: (i) the input of heat from upstream" – maybe you should be a bit more specific here. Mention why you focus on summers. What do you mean by the input of heat from upstream? Tributaries that are warmer than the main stream?"*

Responses: Focus on summer:

We have mainly targeted the biological risk related to global warming.

Introduction § 1.1 line 35 – 37 "As ectotherms, aquatic organisms are very sensitive to ambient water temperature and to its alteration, especially in the vicinity of their upper thermal temperature tolerance (Brett, 1979; Coutant, 1987; McCullough et al., 2009 for Coldwater fish review; Souchon and Tissot, 2012 for European non salmonid fish review).".

*Heat from upstream*

We refer to the conceptual heat flow balance model of Kelleher et al., 2012: the heat flow from upstream depends on the inflow flow Qi and the temperature of the watercourse, which results from the addition of flows from the main river and its tributaries upstream of the studied section.

Kelleher, C., Wagener, T., Gooseff, M., Mcglynn, B., Mcguire, K. and Marshall, L. (2012). Investigating controls on the thermal sensitivity of Pennsylvania streams. Hydrological Processes. 26(5): 771-785.

To be precise we add "fluxes" in L 47

*"Line 50: If you talk about different anthropogenic influences on stream temperature, you probably also should mention cooling water from power plants etc."*

Response: The objective of the study is to quantify the effects small dams in stream; this does not concern cooling water from power plants affecting large rivers.

*"Line 56: > 15 m of what?"*

Fixed
15 m high

*"Line 61 ff: These two "predictions" you are mentioning from 1983 and 1990 should be verified by now? Can you say something about this?"*

The term prediction is inappropriate

Fixed

Previous text: L 61 to 63
In addition, Ward and Stanford (1983) predicted that dams in headwaters might not alter the natural temperature range, with the assumption that canopy and springs or groundwater influx can buffer annual temperature variations.

Replaced by
New text: L64 to 67
In addition, Ward and Stanford (1983) suggest that dams in headwaters might not alter the natural temperature range, with the assumption that canopy and springs or groundwater influx can buffer annual temperature variations. On the other hand, downstream warming may occur during summer releases from surface reservoirs (O'Keeffe et al., 1990).

*"Line 84: With a height smaller than 5m?"*

Fixed
New text: L85
In this work, we studied dams with a height less than 5 m , which we hereafter refer to as small dams.

*"Line 88ff: Be more precise here. There are few articles even considering temperature effects? Those are the 43 sites or articles?"*

These are "studies" in the manuscript of M'Baka et al (2015).
Fixed

*"Line 106: "with closed riparian canopy or aquifers" – what do you want to say here?"*

Previous text: L105 to 106
This variability is greater in headwaters due to the weak thermal inertia and great diversity of these waterbodies, and also to heterogeneous effects with closed riparian canopy or aquifers.

Replaced by

New text: L108 to 110
This variability is greater in headwaters due to the weak thermal inertia and great diversity of these waterbodies, especially with regard to local shading effects from riparian canopy cover and relative importance of spring or tributary discharges.

*"Line 106ff: "This is the reason why it seems preferable in a first study to focus on the single effects of the impoundment immediately downstream the dam." – please reformulate/make your motivation more clear. How exactly is this resulting from the above?"*

Fixed
Previous text L 106 to 107
This is the reason why it seems preferable in a first study to focus on the single effects of the impoundment immediately downstream the dam

Replaced by

New text: L110 to 111
Given this potential complexity with several possible confounding factors, this study focused only on the warming effect of small dams and their impoundment.

*"Line 130: How is a "day of heat wave" defined?"*

*For scenario A1B (mean concentration of greenhouse gases), the estimation was more than ten additional days of heat waves by 2050.*

Response: The definition is conform to International meteorological vocabulary WMO, 1996.
WMO, No. 182. TP. 91. Geneva (Secretariat of the World Meteorological Organization) 1966. Pp. xvi, 276. Sw. fr. 40

"Marked warming of the air, or the invasion of very warm air, over a large area; it usually lasts from a few days to a few weeks"

Fixed

Previous text: L129 to 130
For scenario A1B (mean concentration of greenhouse gases), the estimation was more than ten additional days of heat waves by 2050.

Replaced by

New text: L135 to 136
For scenario A1B (mean concentration of greenhouse gases), the estimation was more than ten additional days of heat waves (WMO, 1966) by 2050.

*"Section 2.2: Mention right away in the text how many dams you study. And how did*
*you chose those specific sites?"*

Fixed

New text: L138
 The 11 dams in the study area are overflow structures and …

The sites were chosen taking into account their distribution in the upstream downstream gradient and the size gradient of the reservoirs.

*Line 145: Make it clear that the temperature sampling was performed for single summers*
*(or two) per site, between 2009 and 2016*

Fixed
We add sentence:
New text: L151 to 153
For two sites, we have data for two different summers (Champagne2009 and 2015, Fretaz  2014 and 2016) because the local water management organization was particularly interested in the thermal regimes of these rivers.  (Table 1).

*"Section 2.5: Please elaborate further on how you performed your PCA. Illustrative variables are*
*explanatory variables? "In order to identify characterization of the impacts of the different dams" –*
*reformulate, unclear!"*

[revised manuscript text omitted]

Figure 7 deleted
The new Figure 7 replaces Figure 8

Figure 7. PCA analysis. Correlation circle with temperature as active variables

[Figure]

Figure 7. PCA analysis. Scatterplot of time series. Ellipses are drawn to visualize the groups obtained with the hierarchical cluster analysis

A new table is added
New text: L602 to 604

Table 4. Results of multiple linear regressions performed on the 2 indicators ΔTmin, ΔTmax using the dam physical characteristics surface area and residence time. Significant p-value are in bold.

| Dependent variable | Independent variable physical characteristics | standardized coefficient | p-value | R2 |
|---|---|---|---|---|
| ΔTmax | surface area | 0.39 | 0.041 | 0.72 |
| | residence time | 0.80 | 0.001 | |
| ΔTmin | surface area | -0.13 | 0.48 | 0.68 |
| | residence time | 0.80 | 0.001 | |

*"Section 3.2/Fig. 4: I understand that the scatter plot for Dompierre shows "type 2", so like in Figure 3. However, Neuf in Fig. 4 does not show "type 1", like in Figure 2, because there is almost no difference between minimum temperatures up- and downstream. And, why don't you simply show the same data in your timeseries plots (Fig.*
*2 and 3) and the scatterplot (Fig. 4) to illustrate the two types. Also, better to combine the figures and make the two types more clear by that."*

Response: We follow the recommendation and propose a new set of figures

Fig. 2 Fretaz 2014 and 2016

[Figure]

and Fig. 4 Dompierre (type 2) and Fretaz (type 1)

[Figure]

Figure 4. Minimum (A) and maximum (B) daily temperatures upstream and downstream of the dams-of-the river (Dompierre site, Veyle stream in 2010; Fretaz site, Veyle stream in 2014). Dashed line is 1:1 line.

Previous text:  L 191 to 194
The two dominant patterns can be illustrated by plotting the minimum and maximum temperature values at the site "Dompierre 2010" with a difference of order of + 1.5°C between the upstream and downstream of the site, comparing to "Neuf 2016", where these values are the same for minimum daily temperatures, or even slightly negative for the maximum temperatures (Fig. 4).

Replaced by
New text: L209 to 212
The two dominant patterns of temperature differences are further illustrated by plotting the minimum and maximum temperature values at the site. For example, at Dompierre in 2010, we observed a consistent shift of approximately +1.5°C (both maximum and minimum daily temperature) between the upstream and downstream of the dam (Fig. 4A). In contrast, at Fretaz in 2014, this shift is dampened, and temperature values between upstream and downstream follow a 1:1 relationship (Fig. 4B).

New figure 4

[Figure]

Figure 4. Minimum (A) and maximum (B) daily temperatures upstream and downstream of the dams-of-the river (Dompierre site, Veyle stream in 2010; Fretaz site, Veyle stream in 2014). Dashed line is 1:1 line.

*"Section 3.3: 0.46% of what?"*

*L 197 This difference averages 0.46% for the 13 cases.*

Response: This precision is deleted, as it is secondary

*"Section 3.5: Specify how you calculate your differences (downstream – upstream?).*
*And don't groups B1 and B2 both exhibit net warming? Be more precise."*

Response: We propose to modify the section 2.4 Data analysis (l 156 à 159)

Previous text:  L156 to 159
To determine if the dams alter the temperature regime, the minimum, average and maximum temperatures and amplitudes were calculated for each full day recorded, and the median values were recorded for the period. The calculations of daily differences of maximum and minimum water temperatures were performed for each pair of upstream/downstream records, and the median of these differences over the recording period was calculated.

Replaced by
New text: L165 to 170
To characterize the influence of dams on stream thermal regimes we first calculated three variables: daily difference between upstream and downstream temperature 1) maximums ($\Delta$Tmax), 2) minimums ($\Delta$Tmin), and 3) amplitudes ($\Delta$Tamp) for each site and year.  With these data, we then conducted the following analyses:
1.      Median summer differences in $\Delta$Tmax, $\Delta$Tmin, and $\Delta$Tamp (median is used instead of mean to limit the influence of extreme values),
2….

*"Section 3.7: Confusing to speak of "short period of time" or "three consecutive days" –*
*what you actually do is to look at shifts in intra-daily temperature variation."*

Fixed

Previous text: L 234 to 239

3.7 Focus on temperature pattern in short period of time
Looking more specifically on a short period of time (three consecutive days), differences in the diurnal variation of the temperature of the river upstream and downstream of the dam shows that for the first group A, the maximum water temperatures upstream and downstream are close, while the minimum temperature downstream does not return to that of upstream (Fig. 9A). In the second group B the water temperature difference between upstream and downstream are more important and remain persistent during all the day period (Fig. 9B).

Replaced by

New text: L244 to 265
3.7  Ecologically relevant intra-daily temperature variation

[revised manuscript text omitted]

*"Section 4, first paragraph: Some of this would be better in the introduction. Same applies to first two paragraphs of section 4.1."*

Response: That's right.
We think that the recall of the context in a few sentences make the discussion as an independently readable part.

*"Line 317, 318: Again, specify the sign of your temperature differences."*

Fixed
New text: L342
with overall median ΔT differences approximately +0.6–2.4°C.

*"Line 344ff: Is Salmo trutta a common species in the rivers of your test sites?"*

Response: Yes, Salmo trutta is endemic and emblematic and at the ecological limit of his distribution. This is why a warming effect added by dams to the natural thermal regime is likely to further limit its range.

*"Line 378: "The thermal landscape is therefore potentially very fragmented due to this fact alone." What do you mean by this and the following sentences?"*

Fixed
Previous text: L378
The thermal landscape is therefore potentially very fragmented due to this fact alone.

Replaced by
New text: L402 to 404
because of the high density of dams in the landscape (0.64 per km), the thermal landscape of this region is potentially fragmented. In other words, we expect that small dams in this region create a discontinuous distribution of stream thermal regimes throughout the river network..

*"Line 385: Please specify which "spatial generalization elements" you mean."*

Fixed
Previous text : L384 to 385
Our work provides spatial generalization elements to better document the present and future thermal landscape

Replaced by

New text: L409 to 410
Our work highlights physical dam characteristics that could be useful in a large-scale heat risk analysis, or in modeling scenarios aiming to account for changes in thermal regimes.

*Technical comments:*
*"Be consistent with thousand separators (for example, you have 2 710000, 96 222, 59071)"*

Fixed

*"Be consistent on how to write "run-of-the-river dam"."*

Fixed

*"Line 38: Why do you cite Rader et al., 2007 as part of the review by Ellis and Jones?"*

Fixed
Previous text:L38
(Rader et al., 2007 in Ellis and Jones, 2013)

Replaced by
New text: L40

(Rader et al., 2007)

*"Line 42: "precipitation", not "precipitations", this comes up several times"*

Fixed
Lines 44,130, 161

*"Line 68: reformulate to "they are expected to increase downstream water temperature"
or similar"*
Fixed
Previous text: L68
they are expected to deliver downstream warmer water

Replaced by
New text: L74
they are expected to increase downstream water temperature

*"Line 78: "(ROE, sept 2017)" why is this cited this way?"*

Fixed
Suppressed

*"Line 59: "water temperature patterns for tens of km"?"*

Fixed
Previous text: L59
alter longitudinal downstream water temperature pattern tens of km

Replaced by
New text: L64
alter longitudinal downstream water temperature pattern for tens of km

*"Line 72ff: "very imprecise depending on national databases. For example, the International
Commission on Large Dams""*

Fixed
Previous text: L 72
nation databases.

Replaced by
New text: L77
national databases.

*"Line 90ff: "Dripps et al. (2013): : :." – please reformulate, sentence unclear"*

Fixed

Previous text : L90 to 92

Dripps et al. (2013) studying 3 residential artificial headwater lakes (17 to 45 ha) on stream (low flow discharge 0.0024 to 0.0109 m3/s) showed that they could increase summer downstream temperature by as much 8.4°C and decrease diurnal variability by as much 3.9°C.

Replaced by

New text: L91 to 93
Dripps et al. (2013) studied the influence of three residential artificial headwater lakes (17–45 ha) on stream (discharge = 0.0024–0.0109 m3 s-1) thermal regimes. They measured a summer downstream temperature increase by as much 8.4°C and a decrease of diurnal variability by as much 3.9°C.

*"Line 95 ff: "Hayes et al. (2008) in the region of the Great Laurentian Lakes" – all this paragraph contains typos and grammar mistakes, please revise"*

Fixed

Previous text: L95 to 97
Hayes et al. (2008) in the region of Great Laurentian Lakes measured a weak to null thermal effect of low-head barriers (<0.5 m in height) built to prevent the upstream migration of sea lamprey Petromyzon marinus, but a temperature elevation comprised between 0.0 to 5.6°C below small hydroelectric dams.

Replaced by

New text: L96 to 99
In the region of Great Laurentian Lakes, Hayes et al. (2008) studied two types of dams with different uses. They measured a weak to null thermal effect of low-head barriers (height <0.5 m) built to prevent upstream migration of sea lamprey (*Petromyzon marinus,* L.). On the other hand, they measured a greater effect for small hydroelectric dams (downstream temperature increases up to 5.6°C).

*"Line 101: Maybe "explaining variables" is a better term"*
Fixed
Previous text:  L 101 to 102
and the difficulty to identify the master variables governing the thermal regime

Replace by

New text: L104
and the difficulty to identify the explaining variables

*"Sector 2.1: Please revise language. Remove repetitive "on a basis of 230 000 km streams with permanent flow""*

Fixed

Previous text: L 121 to 123

with a dam and weir density of 0.64 features per km greater than the French average of 0.42 features per km (Référentiel national des Obstacles à l'Ecoulement, ROE, September 2017) on a basis of 230 000 km for streams with permanent flow.

Replaced by

New text: L127 to 129
Dam and weir density is 0.64 features per km, which is 50% greater than the French average of 0.42 features per km for streams with permanent flow.

Referee #3
*"General comments:*
*The purpose of this study was to quantify the downstream impacts of different types of small dams on summer water temperature in lowland streams. The topic of this manuscript is of high importance, and the research is critically needed since water temperature could impact the structure of aquatic communities and the functioning of the aquatic ecosystem as stated by the authors. The data set on water temperature the authors have collected seems to be robust, and with quite enough number of sites. I personally appreciated the calibration process made for the instruments to insure reliable data. The discussion is quite thorough and insightful, but more focus on literature review (others work) rather than focusing on the discussion of the current work. I found that data analysis severely lacking, and the presentation of the results to be using individual sites as examples that are difficult to judge if they are really representative. Therefore, without adequate data analysis I felt that the conclusions were not well supported. The language used is not sufficiently comprehensible and needs to be improved before publication. Many other specific and technical comments can be found below."*

Response: We have taken all these comments into account and paid particular attention to the statistical analysis of the data to support our conclusions.

*Specific comments*
*"1. P5, L159: Why authors calculate median differences and not mean? Please justifying why this metric instead of means."*

Response: We prefer to work with seasonal variables that are not affected by exceptional one-time weather events.

To avoid any confusion, we eliminate any reference to daily mean temperature
And we propose to modify the section 2.4 Data analysis (l 156 à 159)

[revised manuscript text omitted]

Supressed

*"4. P7, section 3.8: Authors mention that the maximum daily temperature threshold of 22 °C is arbitrary value. While later in the discussion, the authors indicate that the choice of a 22°C  is actually not arbitrary. I suggest that authors delete the word arbitrary and explain the basis of this threshold choice."*

Fixed
Arbitrary is suppressed
See above

*"5. P8, L255: the authors mention warmer, drier, colder and wetter years. Please discuss how these classifications are made?"*

Fixed
Clarification by adding a sentence L 153

New text: L162 to 163
The summer climatic characteristics for our analysis period are compared with the normal values produced by Meteo France (1981–2010).

*"6. P18: Fig.4: what is the reason for comparing temperature of different sites (Dompierre and Neuf) in different years (e.g. 2010 and 2016)."*

Fixed

Response: Figure 4 has been modified. We now use the same sites as in Figures 2 and 3 to make it easier to read. The purpose of the comparison is to illustrate the distribution of the differences in diff_Tmin and diff_Tmax between the two main types of thermal response.
We follow the recommendation and propose a new set of figures (Fig.2 and Fig.4)

Fig. 2 Fretaz  2014 and 2016

[Figure]

and Fig. 4 Dompierre (type 2) and Fretaz (type 1)

Figure 4. Minimum (A) and maximum (B) daily temperatures upstream and downstream of the dams-of-the river (Dompierre site, Veyle stream in 2010; Fretaz site, Veyle stream in 2014). Dashed line is 1:1 line.

Previous text: L 191 to 194

The two dominant patterns can be illustrated by plotting the minimum and maximum temperature values at the site "Dompierre 2010" with a difference of order of + 1.5°C between the upstream and downstream of the site, comparing to "Neuf 2016", where these values are the same for minimum daily temperatures, or even slightly negative for the maximum temperatures (Fig. 4).

Replaced by

New text: L209 to 212
The two dominant patterns of temperature differences can be further illustrated by plotting the minimum and maximum temperature values at the site. For example, at "Dompierre 2010", we observed a consistent shift of approximately + 1.5°C (both maximum and minimum daily temperature) between the upstream and downstream of the dam (Fig. 4A). In contrast, at "Fretaz 2014", this shift is dampened, and temperature values between upstream and downstream follow a 1:1 relationship (Fig. 4B).

*"7. P19: Fig.3 caption: the authors state "time-series of water temperatures upstream (blue line) and downstream (red line) of the dams of Dompierre and Peroux, Veyle stream (2010 and 2015, two warm summer years, respectively + 1.1°C and 2°C, Table 2)", but when looking back in table 2, I have seen that air temperature difference from normal in 2010 is very small (+ 0.3) and NOT +1.1. The +1.1°C air temperature difference from normal is in the year 2009. Therefore, 2009 is almost four times warmer than 2010, hence one may expect the comparison between 2009 and 2015 instead of 2010 and 2015?"*

Fixed
Corrected legend and site changed
Removal of "two warm summer years, respectively + 1.1°C and 2°C, Table 2"  in Fig.3 caption.

*"8. P19: Fig.3: Since air temperature difference from normal in 2010 is very small (+ 0.3), why the difference between upstream and downstream water temperature at Dompierre dam is very high? This cannot be due to long residence time and average surface are in absence of warm condition, so what could be the reason/s?"*

Response: The low deviation from normal indicates a summer temperature close to this normal.
The figure shows that the amount of heat supplied to the stream during a "normal" summer is sufficient to vary the temperature between the upstream and downstream of the dam  taking into account the long residence time (8.4 days) and the surface of the water body (10900 m²).

*"9. It is insecurely to compare 2014 (cold and wet year) with 2015 (warm and dry year) for at least one site (e.g. Dompierre dam) to see the effect of air temperature."*

The difference between the upstream and downstream of the dam does not appear to be solely related to air temperature, as shown in Figure 5.
Unfortunately, we have no data available for the same site for these two years.
We modify Figure 5 and the text as follows:

Previous text L 200 to 204
During the summer season, the differences in the daily mean temperatures upstream / downstream, are close or staggered during all the season. It is notable that the variability of the summer air temperature is much higher (range 17°C) than stream temperature (range 7.5°C) for these examples (Fig. 5), and that the daily water temperature is not well correlated to air temperature.

Replaced by

New text: L217 to 219

During the summer season, the upstream-downstream changes in thermal regime are not well correlated with air temperature for the same periods. For example, a simple linear regression between daily maximum air temperature and $\Delta$Tmax indicates that air temperature explains only 0.8% of the variability in upstream-downstream thermal regime shifts (Fig. 5).

[Figure]

Figure 5. Relation between daily maximum air temperatures (°C), daily upstream/downstream temperature differences for all the data available for the study.

*Technical corrections:*
*"1. P18: in Fig.2 caption, what is the word "respectively" refer to?"*

Fixed
Response: New figure with site Fretaz 2014 – 2016 (Fig.2)
"respectively" is suppressed

*2. P1, L18-19: "The mean increase of the minimum daily temperature was 1°C, with 85 % of the time-series showing an increase > 0.5 °C", this sentence is not clear or grammatically incorrect.*
Fixed
Previous text: L18 to 19
The mean increase of the minimum daily temperature was 1°C, with 85 % of the time-series showing an increase > 0.5 °C.

Replaced by

New text: L18 to 19
Across all time series, the mean increase of $T_{min}$ was 1°C. For 85% of time series, the increase in $T_{min}$ was greater than 0.5°C.

*"3. P2, L63-64: "surface release reservoirs", should read "surface reservoirs' release"."*

Fixed

*"4. P5, L148-149: "in the main flow of the channel" should read "in the main flow channel"."*

Fixed

*"5. P5, L151: "method Dunham et al. (2005)." should read "method introduced by Dunham et al. (2005)"."*

Fixed

*"6. P5, L157: the authors state that "and the median values were recorded for the period", how do you record the median? It should read "calculated" instead."*
Fixed

*"7. P6, L182: "Furthermore, the average temperature downstream of the structure was systematically higher or equivalent than that measured upstream" should read "Furthermore, the average temperature downstream of the structure was systematically equivalent or higher than that measured upstream"."*
Fixed

*"8. These are limited examples and the paper contains more. All grammatical errors should be fixed before publication."*

A final revision of English was done by a native speaker

**List of all relevant changes made in the manuscript,**

The major changes we have made are:
- use of statistical analysis methods to strengthen the robustness of the results,
- improved consistency between points raised in the text and proposed figures,
- grammatical quality review.

Material and methods (section 2)

Temperature data analysis (Section 2.4)
This section has been completely rewritten to allow a more fluid reading, and to eliminate ambiguities: calculations of daily differences, choice of the seasonal median, and calculations of variables assessing the temperature exceeding 22° C.

Hierarchical cluster analysis (HCA) to classify the time series and results (new section 2.5)
Use of the HCA to classify time series thermal regime groups.

Ordination analysis (section 2.6 instead of PCA analysis)
Use of the ordination analysis, redundancy analysis and multiple linear regression to identify and to determine specific effect sizes of the relevant physical dam characteristics.

Results (section 3)

General temperature pattern (section 3.1) and magnitude of upstream/downstream differences (section 3.2)
We have chosen a more appropriate example (Fretaz site for summer 2014, summer 2016) to illustrate the main types of effects of weirs on water temperature (new Fig. 2 and Fig.3)

Correlation with air temperature (section 3.4)
A new linear regression between daily maximum air temperature and daily maximum water temperature differences show the weak correlation between these two variables.

Results of statistical analysis: (section 3.5 Site typology and section 3.6 Ordination results)
Comments on the result of statistical calculations that confirm the results presented in the initial manuscript by giving them numerical orders of magnitude.

Temperature pattern in intra daily variation (Section 3.7)
We have completed this section by adding the calculation result of the average daily duration exceeding 22°C upstream and downstream the dam. A new figure is added (Fig. 11) and belonging to groups A, B1 and B2 is indicated on the two figures.

Discussion (section 4)

The whole text has been improved to make it more fluent to read.

**Marked-up manuscript version**

All changes concerning vocabulary, grammar and typography are not marked in this document.

Only changes that have occured after the discussion phase and those that bring new elements to the manuscript are marked-up.

This concerns the main text, bibliography, tables and figures

**Determinants of thermal regime influence of small dams**

André Chandesris [1], Kris Van Looy [2], Jake Diamond[1], Yves Souchon [1]

[revised manuscript text omitted]

**Commentaire [CA24]:** in response to comments from referee #3

clarification requested about warmer, drier, colder and wetter years

**Commentaire [CA25]:** in response to comments from referee #2

clarification requested about calculation of the differences (downstream-upstream)

**Commentaire [CA26]:** response comments referees #1; #2; #3 request to clarify section 3.7

sentence added to explain calculations about thermal threshold 22 °C and potential biological effects.

**Commentaire [CA27]:** in response to comments from referee #1 and #3

clarification about mean vs. median

**Commentaire [CA28]:** in response to comments from referee #1 and #3

clarification about statistical method for group clustering : hierarchical cluster analysis (HCA)

**Commentaire [CA29]:** in response to comments from referee #1 and #2

clarification about the PCA analysis and explanatory variables

**Results**

**General temperature patterns**

Regardless of site or year, we observed consistent a pattern of summer temperature variations consisting of the following (Fig. 2):

- daily (diel) variation (minimum in early morning, maximum in late evening),
- periods of progressively increasing $T_{min}$ and $T_{max}$, and
- rapid drops in temperature that interrupt these periods, and that are generally linked to precipitation events.

The periods of progressively increasing temperature vary in length, magnitude, and timing from one year to another, but the general pattern remains the same, as demonstrated by the case of the Fretaz dam , monitored in 2014 (a cold and humid year) and 2016 (a more normal year, Fig. 2; Table 2).

We observed two consistent patterns in upstream-downstream thermal regimes. In the first pattern, $T_{min}$ is higher downstream, but $T_{max}$ stays relatively constant (Fig. 2). We note that these upstream-downstream differences were muted in 2014, the cold and humid year (Fig. 2). This thermal pattern (i.e., where $T_{min}$ increases downstream, but not $T_{max}$) is observed in 7 out of 13 cases (Table 3). In the other cases (6 out of 13; Table 3), we observed a second pattern, where both $T_{min}$ and $T_{max}$ are higher downstream of the structure, which results in a consistent shift between the two temperature time-series (Fig. 3, selected examples: Dompierre dam 2010 and Peroux dam 2015).

**Magnitude of upstream-downstream differences**

The two dominant patterns of temperature differences are further illustrated by plotting downstream versus upstream $T_{min}$ and $T_{max}$ values at the site. For example, at Dompierre in 2010, we observed a consistent shift of approximately +1.5°C (both $T_{min}$ and $T_{max}$) between the upstream and downstream of the dam (Fig. 4A). In contrast, at Fretaz in 2014, this shift is dampened, and temperature values between upstream and downstream more closely follow a 1:1 relationship (Fig. 4B).

**Reduction in the daily amplitude of downstream temperatures compared to upstream temperatures**

We also observed that $\Delta T_{amp}$ was reduced for 61.5% of our time series (Table 3). This reduction in amplitude is primarily due to a truncated daily minimum downstream temperature that is on average 0.96°C higher than that of the upstream.

**Dam thermal effects are not correlated with air temperature**

During the summer season, the upstream-downstream changes in thermal regime are not well correlated with air temperature for the same periods. For example, a simple linear regression between daily maximum air temperature and $\Delta T_{max}$ indicates that air temperature explains only 0.8% of the variability in upstream-downstream thermal regime shifts (Fig. 5).

**Site typology**

The hierarchical cluster analysis applied to the daily summer temperature anomalies distinguished three groups:

**Commentaire [CA30]:** in response to comments from referee #1

clarification request the choice of the example showed in Figure 2
(site changed)

**Commentaire [CA31]:** in response to comments from referee #1

more explanation about uselfulness of Figure 3

**Commentaire [CA32]:** in response to comments from referee #1; #2 and #3

about section 3.2 and the link with the Figure 4
Site is changed according to the previous section 3.1 and figures 2 and 3

**Commentaire [CA33]:** in response to comments from referee #1 and #3

correlation with air temperature by plotting daily data across all sites

- a first group (A) characterized by:
    - median of $\Delta T_{max}$ less than 0.5°C;
    - median of $\Delta T_{min}$ between + 0.4–1.3°C;
    - median of $\Delta T_{amp}$ less than -0.2°C.
- a second group (B1) characterized by:
    - median of $\Delta T_{max}$ ranging from +0.6–1.2 °C;
    - median of $\Delta T_{min}$ between +0.3–1.1°C.
- a third group (B2) characterized by:
    - median of $\Delta T_{max}$ greater than 1.2 °C;
    - median of $\Delta T_{min}$ greater than 1.2 °C

The distribution of the differences between the minimum and maximum temperature values during summer (Fig. 6) confirms the difference between these three groups.

**Ordination **analysis**

The first axis of the PCA analysis (74.1% of total inertia) is correlated to all daily temperature daily anomalies, in particular to the $\Delta T_{max}$. The second axis (25.3%) discriminates the $\Delta T_{amp}$ with $\Delta T_{min}$ (Fig. 7). Results of the RDA show that the water residence time and the impoundment surface explain 95.2% of the PCA structure. The projection of the sites on these axes shows a strong spreading along the first axis (Fig. 7). Additionally, the dams that had two different measurement years stay within the same range on this first axis (i.e., Fretaz and Champagne) (Fig. 7).

Multiple regression analyses between the temperature variables (median values of $\Delta T_{min}$ and $\Delta T_{max}$) and the physical characteristics obtained by the RDA (residence time and impoundment surface) resulted in high explanatory power ($R^2 \approx 0.7$). These regressions identified the significant contribution of residence time for $\Delta T_{min}$ and $\Delta T_{max}$, whereas only surface area had a significant contribution for $\Delta T_{max}$ (Table 4).

**Ecologically relevant intra-daily temperature variations**

To further illustrate the different thermal regime effects from our typology analysis, we compare intra-daily temperature variations for a three-day time series in group A (small thermal effect) with group B (large thermal effect; Fig. 8):
- In the example of group A (Fig. 8A), the downstream temperature is generally warmer than the upstream temperature (observed difference of 1°C warmer) except for a few hours during the three day sample observation period. The biological benchmark of 22°C is exceeded both upstream and downstream during the day of August 20. The rest of the time, temperatures are below this threshold. From a biological point of view, the duration above the thermal threshold is short, preceded and followed by more favorable temperatures (i.e., the remission period).
- In the example of group B (Fig. 8B), the downstream temperature is systematically higher than that of the upstream, with a temperature difference varying between +0.8–2.4°C. The 22°C threshold is exceeded downstream for a cumulative 42 h over the three-day period. August 15 and 16 have downstream temperatures that rarely go below 22°C, leaving no time for thermal remission (return to a

**Commentaire [CA34]:** in response to comments from referee #1 and #3

results of statistical method for group clustering (HCA)

**Commentaire [CA35]:** in response to comments from referee #1 and #2 clarification requested about the PCA analysis

results of new Ordination analysis and the explanatory variables highlighted by the redundancy analysis

[revised manuscript text omitted]

**Commentaire [CA38]:** in response to comments from referee #2

discussion about most important drivers of temperature regime (surface and time residence) confirmed with multiple analysis regression

**Commentaire [CA39]:** in response to comments from referee #2

complements about notion of the possibility of regionalization

[revised manuscript text omitted]

**Commentaire [CA41]:** response comments referees #2

clarification about "fragmented thermal lanscape "

**Commentaire [CA42]:** response comments referees #2

clarification about "spatial generalization elements"

[revised manuscript text omitted]

**Commentaire [CA45]:** added reference for stiscal method used in the new version

**Commentaire [CA46]:** reference added in response to referee #2 comment about transferability of results

**Commentaire [CA47]:** reference added for "day of heat wave" definition

**Table 1. Physical characteristics of dams of the river and impoundments.**

[revised manuscript text omitted]

**Commentaire [CA51]:** figure changed according to in response to comments from referee #1 and #3

correlation with air temperature by plotting daily data across all sites

[Figure]

**Figure 6. Box-plot distribution (25% - 75 %) of upstream/downstream differences of daily maximum and minimum temperatures for all the time series studied. (Red lines: 0°C for daily maximum temperature and 1°C for daily minimum temperature are drawn to help reading). The vertical lines drawn in bold are the limits to the three classes of results of the hierarchical cluster analysis. Dendrogram CAH's result is shown at the top left of the figure.**

**Commentaire [CA52]:** change of the figure according to results of HCA (section 3.5)

[Figure]

**Figure 7. PCA analysis. Scatterplot of time-series. Ellipses are drawn to visualize the groups obtained with the hierarchical cluster analysis**

**Commentaire [CA53]:** change according to new results of ordination analysis (section 3.6)

[Figure]

**Figure 8. Time-series of water temperatures upstream (blue line) and downstream (red line) of the dams of A/ Caillou (Vieux Jonc stream) and B/ Revel (Solnan stream) focused on three days during August.**

[Figure]

**Figure 9 Mean of the daily maximum duration with T above 22 °C , upstream and downstream each site monitored in the study. A (circles), B1 (triangles), B2 (rhombus) are the groups of sites resulting from HCA.**

**Commentaire [CA54]:** change according to results of HCA (site typology)

[Figure]

**Figure 10. Percentage of number of summer days with a diurnal maximum temperature of water greater than 22 °C, upstream and downstream each site monitored in the study. A (circles), B1 (triangles), B2 (rhombus) are the groups of sites resulting from HCA.**

**Commentaire [CA55]:** new figure added to highlight links between site typology and biological potential effects.

---

## Author Response (AR2)

**Point-by-point response to the review**

*Manuscript title:* "Small dams alter thermal regimes of downstream water"

*Initial submitted:* 04 Apr 2019

*Major revision submitted:* 22 Jul 2019

*Status:* minor revision

Dear Editor and Referees,

Thank you for the quality of your proofreading and comments. We also appreciate your interest in the subject matter. We believe we have addressed all of the last outstanding comments and issues.

Below we respond in normal text to *comments in italics*, while showing old text in blue, and replacement text in green. Here, we note that we have changed the title of our manuscript to improve its clarity.

*Line 28: "that these effects can be accurately predicted" – these are bold words, given the limited variability of sites and environmental conditions within your dataset, and the limited number of explanatory variables you actually tested.*

We will temper our language here, but disagree with the reviewer that the limitations of our analysis preclude the general statement. We considered what we believe are a sufficient number of explanatory variables that are applicable to small dams throughout the region (n = 8, residence time, surface area, Strahler order, distance to the source, watershed area, dam height, impoundment length, and impoundment volume. The statistical analysis (redundancy analysis with automatic stepwise selection) identifies only two significant predictors (surface area and residence time) with a coefficient of determination = 0.7, which we deem as sufficiently accurate intent of our statement. We suggest that this result clearly shows the importance of these two variables relative to the many other variables considered. Therefore, we concluded that these two variables can provide water managers with sufficient accuracy to prioritize restoration projects.

Following this observation, we propose to change the sentence.

Line 28 "...that these effects can be accurately predicted with two simple measurements of small dam physical attributes."

35 Line 28 "...that these effects can be predicted with sufficient accuracy ($R^2$=0.7) with two simple measurements of small dam physical attributes."

*Line 194 ff: I would also characterize a fourth pattern in the temperature development, which is a period of progressively decreasing Tmin and Tmax (but not as a sharp drop), e.g. from 13/8/2014 to 18/8/2014 in Figure 2. Such a characteristic likely is related to relatively cool weather, maybe with moderate precipitation, or what do you assume?*

We appreciate the reviewer's attention to detail in this figure, but note here that this observed pattern only occurs at one particular instance. The particular case of this cooling with a gradual decrease in $T_{max}$ and $T_{min}$ in August 2014 corresponds to a period of moderate precipitation with a cloudy weather for several days following an episode of intense precipitation (40 mm /24hrs). This type of weather situation, observed in 2014, a rather cold and humid year (Table 2) was not observed in the other years, so we do not mention it in the list of summer-to-summer in the study area.

*Section 3.4 / Figure 5: I do not quite understand why you talk about correlation between air temperature and the upstream-downstream difference in stream temperature. It is not strong; I assume you have thought of why you looked into it (besides it being requested by a reviewer), but it does not become clear to the reader why. Either explain this, or maybe leave out Figure 5 as it does not seem to add much information, and just mention the results of a simple correlation analysis.*

We fully agree to remove Figure 5, which does not provide much information. We have also condensed those short sections (3.2–3.4) into one section, and explained in-text the results of the correlation analyses and that the purpose behind it was to test for daily radiative influence on observed upstream-downstream differences.

*Figure 6: The cluster dendrogram should relate to your group names. I assume that gp 12-13 is group B2, group 8-11 is group B1, and gp 1-7 is group A. However, if this is the case, why do you refer to the two more distinct groups (according to your cluster analysis) with names B1 and B2? Also, intuitively, group B2 is most distinct, whereas B1 and A are more similar, as only B2 has both uniquely high delta Tmin and delta Tmax. Your distinction probably originates from group A being considered less problematic ecologically (low increase in Tmax), whereas both group B1 and B2 show a higher increase in Tmax, but this choice should be discussed, also wrt. the cluster analysis.*

We thank the reviewer for noticing this error, and we have edited the Figure accordingly. With regards to the comment about groups A, B1, and B2, we note that although group B2 appears at first glance to be very different from both groups B1 and A (suggesting that it should be a separate group entirely), the difference between A and

B is the most ecologically distinct. In group A, only $\Delta T_{min}$ is greater than zero. In groups B, $\Delta T_{min}$ and $\Delta T_{max}$ are both greater than zero, which has the most impact on the functioning of aquatic organisms. We maintain our original groupings, but provide the following additional explanatory text.

New line 224 "The hierarchical cluster analysis differentiates the B2 group primarily from the B1 and A groups (Fig. 6). We propose to retain the major distinction between group A and group B, because it is based on a temperature increase between upstream and downstream, only for $T_{min}$ (group A), but for $T_{min}$ and $T_{max}$ (group B), which is an important threshold for the physiology of aquatic organisms."

*Line 245 ff: For my understanding, you describe the two different timeseries in an unnecessarily complicated manner. Isn't the whole point of your distinction between group A and B, that group A does NOT show a siginifcant/consistent increase in Tmax, whereas group B does? Then I suggest mentioning this directly instead of saying that "the downstream temperature is generally warmer than the upstream temperature [...] except for a few hours". So, the point is not only that "the duration above the thermal threshold is short" (line 250 f.), but more that group A does not show a consistent increase in Tmax, which usually is the ecologically problematic temperature?*

We agree that the writing was overly complicated, so we have simplified the idea, focusing on the primary upstream-downstream differences and the duration of threshold exceedance. We have also added a horizontal line on Figure 7 at 22 degrees to illustrate the threshold exceedance.

Line 245 "In the example of group A (Fig. 8A), the downstream temperature is generally warmer than the upstream temperature (observed difference of 1°C warmer) except for a few hours during the three day sample observation period. The biological benchmark of 22°C is exceeded both upstream and downstream during the day of August 20. The rest of the time, temperatures are below this threshold. From a biological point of view, the duration above the thermal threshold is short, preceded and followed by more favorable temperatures (i.e., the remission period)."

New line 244 "In the example of group A (Fig. 7A), the downstream thermal warming effect is limited to the nighttime ($T_{min}$) period (observed difference of 1°C warmer). Additionally, although the biological benchmark of 22°C is exceeded both upstream and downstream during the day of August 20,the duration above the thermal threshold is short, preceded and followed by more favorable temperatures (i.e., the remission period)."

*Line 257 ff: This is hard to understand, please reformulate. Plus, can you state this based on only the three days from two example sites?*

105

The purpose of this paragraph is to show at the daily time scale the difference between a persistent temperature shift between upstream and downstream (group B) and a less significant variation in temperature differences allowing organisms to recover below a thermal threshold.

This figure simply zooms in on the different time series whose groups are identified by cluster hierarchical analysis.

110    We have clarified the text as requested.

Line 257 "Additionally; differences in the diurnal temperature variation upstream and downstream of the dam shows that for group A, the maximum water temperatures are similar, whereas the minimum temperature downstream does not return to that of upstream (Fig. 8A). In group B the water temperature difference between

115    upstream and downstream are persistent throughout the diurnal cycle (Fig. 8B)."

New line 248 "These differences between the downstream responses in diurnal temperature variation hold throughout the time series. In other words, group A has a consistent response of no change in downstream maximum water temperatures, coupled to a consistent increase in downstream minimum temperature (e.g., Fig.

120    7A). Group B differs in that the downstream maximum temperatures are also increased (e.g., Fig. 7B)."

*Line 330ff: "Group A is characterised by a residence time less than 0.7 days [...] In group A, we suggest that long residence times reduce cooling effects" – isn't that formulated confusingly? You mean, that even the short (as in shorter than most in group B) residence time already reduces the nocturnal cooling effect? And why should the*

125    *large dams from group B with a short residence time (0.2 days) still experience heating from solar radiation? Shouldn't that be mainly related to the residence time (hence, many of the smaller dams should also experience heating from solar radiation? Or are the smaller dams shaded by vegetation to a larger extent than the larger ones? If so, canopy cover/shading of the reservoir areas should really be considered as another explanatory variable. I assume you do not really know this, but at least try to discuss some of the potentially counter-intuitive*

130    *results).*

This was a typo on our part (Group A should have been Group B), which we believe contributed to much of the confusion from the reviewer. We have fixed this error and added some clarifying text about the differences between the two B groups.

135

Line 330  to 336"For example, Group A is characterised by a residence time less than 0.7 days and an impoundment surface area smaller than 35,500 m², whereas group B is characterised either by a long residence time (e.g., Dompierre dam with residence time = 8.4 days and surface area = 10,900 m²), or by a surface area larger than 35,000 m² with a shorter residence time (e.g., 0.2 days). These physical differences are directly linked

140 to the observed differences in thermal regime shifts. In group A, we suggest that long residence times reduce cooling effects; the nocturnal input (i.e., the cooling effect) becomes negligible in the general heat exchange balance. Group B also exhibits this reduced cooling effect, but exhibits an additional heating effect linked to increased solar radiation from larger impoundment surface areas."

145 New line 324 to 330 "For example, Group A is characterised by a short residence time (less than 0.7 days) and a small impoundment surface area (less than 35,500 m²), whereas group B is characterised either by a large surface area (greater than 35,000 m²) with a short residence time (e.g., 0.2 days; group B1), or by long residence times (e.g., 8.4 days; group B2,). These physical differences are directly linked to the observed differences in thermal regime shifts. In group B2, we suggest that long residence times reduce cooling effects; the nocturnal input (i.e.,
150 the cooling effect) becomes negligible in the general heat exchange balance. However, for group B1 dams with short residence time, but large surface areas, increased energy supply by solar radiation on the larger surface may overwhelm any potential cooling effects."

*Line 410: "For example, a simple model using only small dam residence time and surface area may be able […]"*
155 *Isn't this a bold statement, given what I mentioned in the above paragraph regarding line 330 ff.?*

We agree with the reviewer that this may indeed a bold statement if applied to a particular case study. However, we believe that our language holds when considering the results of this study at a more regional scale, as implied in our statement. It is necessary for land and water resource managers to be able to identify simply and quickly
160 the impoundments in greatest need for restoration. We suggest that the metrics identified here can be used with sufficient accuracy to be able to identify such impoundments. Hence, we have adjusted the language to more precisely state our intention.

Line 410 "For example, a simple model using only small dam residence time and surface area may be able to
165 predict with sufficient accuracy thermal regime change at the regional scale."

New line 403 "For example, a simple model using only small dam residence time and surface area may be able to diagnose with sufficient accuracy thermal regime change at the regional scale."

**Small dams alter thermal regimes of downstream water**

André Chandesris [1], Kris Van Looy [2],  Jacob S. Diamond[1], Yves Souchon [1]

[1] River Hydro-Ecology Lab, National Research Institute of Science and Technology for Environment and Agriculture, UR Riverly, Lyon, France

[2] OVAM, Stationsstraat 110, 2800 Mechelen, Belgium

*Correspondence to*: A. Chandesris (andre.chandesris@irstea.fr)

**Abstract.**

The purpose of this study was to quantify the downstream impacts of different types of small dams on summer water temperature in lowland streams. We examined: 1) temperature regimes upstream and downstream of dams with different structural characteristics, 2) relationships between stream temperature anomalies and climatic variables, watershed area, dam height, impoundment length and surface area, and residence time,  3) the most significant variables explaining the different thermal behaviours, and 4) the dam thermal effect considering a biological threshold of 22 °C, with a calculation of both the number of days with temperature above this threshold and the average hourly duration above this threshold.

Water temperature loggers were installed upstream and downstream of 11 dams in the Bresse Region (France) and monitored at 30 min intervals during summer (June to September) over the period 2009–2016, resulting in 13 paired water temperature time-series (two sites were monitored for two summers, allowing the opportunity to compare cold and hot summers).

At 23% of the dams, we observed increased downstream maximum daily temperatures by greater than 1°C; at the remaining dams we observed changes in maximum daily temperature -1–1°C. Across sites, the mean downstream increase of the minimum daily temperature was 1°C, and for 85% of the sites this increase was higher than 0.5°C.

We  hierarchically clustered the sites  based on  temperature anomaly variables  upstream-downstream differences in 1) maximum daily temperature ($\Delta T_{max}$), 2) minimum daily temperature ($\Delta T_{min}$), and 3) daily temperature amplitude ($\Delta T_{amp}$).  The cluster analysis identified two main types of dam effects on thermal regime: 1) a downstream increase in $T_{min}$ associated with $T_{max}$ either unchanged or slightly reduced for impounds of low volume (i.e., residence time shorter than 0.7 day and surface area less than 35,000 m$^2$), and 2) a downstream increase of both $T_{min}$ and $T_{max}$ on the same order of magnitude for impounds of larger volume (i.e., residence time longer than 0.7 day and surface area greater than 35,000 m$^2$). These downstream temperature increases reached 2.4°C at certain structures with the potential to impair the structure of aquatic communities and the functioning of the aquatic ecosystem.

Overall, we show that small dams can meaningfully alter the thermal regimes of flowing waters, and that these effects can be accurately predicted with two simple measurements of small dam physical attributes. This finding may have importance for modelers and managers who desire to understand and restore the fragmented thermalscapes of river networks.

**Keywords:** Stream, water temperature, impoundment, weir, run-of-river dam

[revised manuscript text omitted]